# Development of the social brain from age three to twelve years

Hilary Richardson [1], Grace Lisandrelli[1], Alexa Riobueno-Naylor[2] & Rebecca Saxe[1]

Human adults recruit distinct networks of brain regions to think about the bodies and minds of others. This study characterizes the development of these networks, and tests for relationships between neural development and behavioral changes in reasoning about others' minds ('theory of mind', ToM). A large sample of children ($n = 122$, 3–12 years), and adults ($n = 33$), watched a short movie while undergoing fMRI. The movie highlights the characters' bodily sensations (often pain) and mental states (beliefs, desires, emotions), and is a feasible experiment for young children. Here we report three main findings: (1) ToM and pain networks are functionally distinct by age 3 years, (2) functional specialization increases throughout childhood, and (3) functional maturity of each network is related to increasingly anti-correlated responses between the networks. Furthermore, the most studied milestone in ToM development, passing explicit false-belief tasks, does not correspond to discontinuities in the development of the social brain.

[1] Department of Brain and Cognitive Sciences, Massachusetts Institute of Technology, Cambridge, MA 02139, USA. [2] Department of Psychology, Wellesley College, Wellesley, MA 02481, USA. Correspondence and requests for materials should be addressed to H.R. (email: hlrich@mit.edu)

Over the past decade, fMRI research has made significant progress identifying functional divisions of labor within the adult social brain[1]. For example, while many areas of human cortex show elevated responses while looking at, listening to, or thinking about other people, studies of these cortical responses suggest a striking division between regions responding preferentially to internal states of others' bodies, versus internal states of others' minds[2–6]. Both bodily sensations, like hunger and pain, and mental states, like beliefs and desires, are internal states of other people; both are important for observers' reasoning about others' actions and reactions, to facilitate the observer's own prosocial (e.g., helping) or antisocial (e.g., competing) choices. In spite of these similarities, a robust dissociation between responses to others' bodies and minds has been replicated across a wide range of paradigms: when human adults think about other people, our cortical responses are surprisingly dualist[7].

An important extension of this work is to study the emergence of these functionally specialized brain regions during development. The current study investigates the developmental origins of the cortical dissociation between others' bodies and minds, and the links between cortical and cognitive changes in children's social development.

Although children's developing understanding of others' minds (their 'theory of mind' (ToM)) has been studied intensively[8], we know very little about the neural changes that support this development. One cause of this gap in knowledge is that most behavioral studies on ToM focus on children younger than 5 years old[9,10]. For example, one active debate in developmental psychology concerns children and infants' ability to reason about false beliefs[11]. Children's ability to explicitly predict or explain another person's actions based on her false beliefs has been interpreted as depending on a conceptual leap occurring around age 4 years[12–14]. However, recent measures of spontaneous looking and helping suggest that even toddlers may be sensitive to others' false beliefs[15,16]. By contrast, fMRI studies of ToM reasoning have focused on children older than 5 years old[17–23], adolescents[24,25], and adults[26–28]. Prior neuroimaging studies thus leave open questions of core interest concerning early stages of theory of mind development.

Based on theories in developmental psychology, we derive three predictions for observations in the social brain regions of young children. First, success on explicit false-belief tasks could reflect an important conceptual leap or discontinuity in ToM development, as theories of others' internal states are dramatically altered by insight into the representational nature of mental states[29,30]. According to this view, the division between cortical responses to others' bodies versus minds might emerge concurrently with childrens' explicit understanding of false beliefs. Second, success on explicit false-belief tasks could reflect development in other domain-general brain regions, removing earlier performance limitations (such as response inhibition and selection, and production of verbal response)[31–33]. According to this view, spontaneous processing of others' mental states within domain-specific regions for ToM might be similar in children who pass and fail explicit false-belief tasks. Third, success on explicit false-belief tasks could be a single step in the ongoing conceptual development of ToM, which begins before—and continues after–false-belief reasoning[34–37]. According to this view, change within ToM brain regions might occur both before and after children explicitly reason about false-beliefs. Of course, these predictions only reflect a subset of those that could be derived from each theoretical perspective, and are not mutually exclusive; reality could include a mixture of these three views.

The present study characterizes development of brain regions recruited for reasoning about others' minds and bodies, in a large, cross-sectional sample of children between the ages of 3–12 years

old. These 122 children and a reference group of 33 adults, watched a short, animated movie that included events evoking the mental states and physical sensations of the characters, while undergoing fMRI. Watching this movie is feasible for young children—it is short, engaging, and does not require learning a task. This movie has been validated as activating ToM brain regions and the pain matrix in adults[38]. ToM brain regions include bilateral temporoparietal junction, precuneus, and dorso-, middle-, and ventromedial prefrontal cortex[26–28]. The pain matrix includes brain regions recruited when perceiving the physical pain and bodily sensations of others: bilateral medial frontal gyrus, insula, and secondary sensory cortex, and dorsal anterior middle cingulate cortex[39]. Within both functional networks, individual regions have been implicated with specific functions (for example, insula and cingulate cortex for nociceptive pain[39], and prefrontal cortex for reasoning about emotions and preferences[40]). Here, we collapse across specific functions, and operationalize ToM and pain networks as regions recruited generally for reasoning about others' internal mental and physical states, respectively[38].

We measured three features of children's hemodynamic responses during the movie. First, we conducted inter-region correlation analyses to test the degree to which ToM and pain brain regions operate as functionally distinct networks (i.e., high within-network, and low between-network correlations)[41,42]. Because results suggested that networks for ToM and pain are distinct even in the youngest children, we used the average response of each network in the next two analyses. Second, we measured the magnitude of evoked response, in children, to the events in the movie that evoke peak responses in adults (identified by reverse correlation analyses). Third, we measured the functional maturity (i.e., similarity to adults) of each network's entire timecourse[43]. All child participants additionally completed an assessment of explicit ToM after the scan, to measure overall theory of mind reasoning, including performance on explicit false-belief tasks. We tested whether each of the three neural measures was related to children's age, to children's explicit performance on ToM tasks, and to one another.

We report evidence that ToM and pain networks are functionally distinct by 3 years of age, and become increasingly specialized between the ages of 3–12 years. Functional maturity of each network is related to increasingly anti-correlated responses between the two networks. Finally, we find that a distinct neural response to others' minds and bodies is present before—and continues to develop after—children pass explicit false-belief tasks.

## Results

**Behavioral results.** All children completed a behavioral battery after completing the fMRI scan, which included a custom-made explicit ToM task (see Methods)[21]. 3- to 5-year-old children ($n = 65$) additionally completed a measure of response inhibition (Dimensional Change Card Sort task (DCCS)[44]). Performance on the ToM task (proportion correct) and DCCS were both positively correlated with age (ToM (kendall tau correlation test ($n = 122$)): $r_k(120) = .66$, $p < .00001$; DCCS (kendall tau correlation test ($n = 64$)): $r_k(62) = .20$, $p = .049$; see Fig. 1a. In the 3-year to 5-year-old subset of children who completed both measures, ToM and DCCS scores were positively correlated (partial kendall tau correlation test ($n = 64$), controlling for age: $r_k(61) = .19$, $p = .03$). See Supplementary Table 1 for behavioral data and participant demographics.

For 3- to 5-year-old children, an explicit false-belief composite score was calculated based on responses to six explicit false-belief questions embedded within the ToM measure; this composite

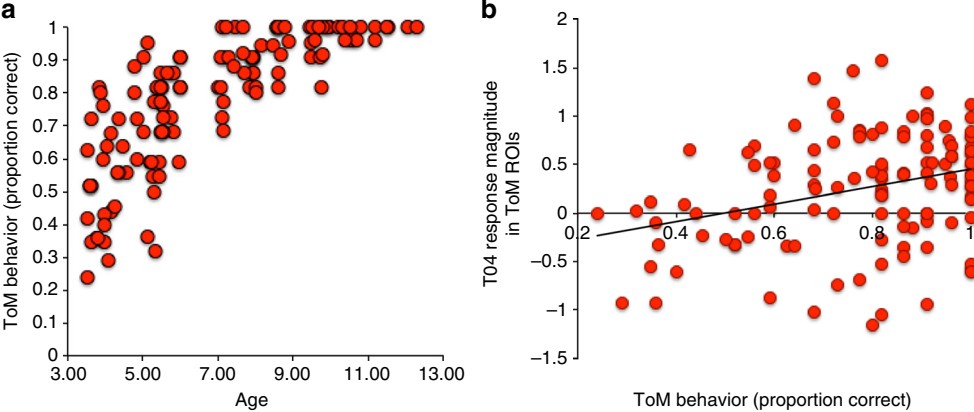

**Fig. 1** Theory of mind behavioral performance. **a** Theory of mind behavioral performance (proportion correct; *y*axis) of all children (*n* = 122) by age in years (*x*axis). **b** Average response magnitude in ToM network to peak timepoint of event T04 (Peck returning to Gus, donning protective gear), per child (*y*axis), by theory of mind behavioral performance (proportion correct; *x*axis)

measure was used to categorize these children as false-belief passers (5–6 FB questions correct; $n = 30$ (15 female)), inconsistent performers (3–4 FB questions correct; $n = 20$ (13 female)), and false-belief failers (0–2 FB questions correct; $n = 15$ (6 female)). False-belief task failers and inconsistent performers did worse on the remaining ToM items than passers (Fail $M$(s.e.) $= .55(.04)$, Inc $M$(s.e.) $= .57(.03)$, Pass $M$(s.e.) $= .75(.02)$; Tukey Honest Significant Difference (HSD) test of ToM*FB-Group ANOVA: Pass-Fail: diff $= 1.2$, $p < .00005$; Pass-Inc: diff $= 1.08$, $p < .0001$; Inc-Fail: diff $= .16$, $p = .8$; Kruskal–Wallis rank sum test of ToM*FB-Group (for non-normal distributions; 3 groups: Pass ($n = 30$), Inc ($n = 20$), Fail ($n = 15$)): $H(2) = 22.96$, $p < .0001$). False-belief task failers were on average younger than passers and inconsistent performers (Fail $M$(s.d.) $= 4.1(.56)$ years; Inc $M$(s.d.) $= 4.8(.73)$ years; Pass $M$(s.d.) $= 5.2(.70)$ years; Tukey HSD test of Age*FB-Group ANOVA: Pass–Fail: diff $= 1.4$, $p < .00001$; Inc-Fail: diff $= .83$, $p = .01$; Pass-Inc: diff $= .59$, $p = .047$). Similarly, failers demonstrated worse response inhibition than the other two groups (DCCS Summary score: Fail $M$(s.e.) $= 1.73(.21)$, Inc $M$(s.e.) $= 2.26(.17)$, Pass $M$(s.e.) $= 2.33(.09)$; Tukey HSD test of DCCS*FB-Group ANOVA: Pass–Fail: diff $= .88$, $p = .01$; Inc-Fail: diff $= .78$, p $= .052$; Pass-Inc: diff $= .1$, $p = .9$; Kruskal–Wallis rank sum test of DCCS*FB-Group (for non-normal distributions; 3 groups: Pass ($n = 30$), Inc ($n = 19$), Fail ($n = 15$)): $H(2) = 7.56$, $p = .02$).

**Inter-region correlation analysis**. Inter-region correlation analyses reveal the extent to which a group of brain regions operate as a network with synchronized responses. We conducted inter-region correlation analyses (see Methods)[42], in order to test three hypotheses about the development of ToM and pain brain regions: (1) that adults exhibit greater within-network correlations and greater anti-correlations between ToM and pain networks, compared to children, (2) that by age 3 years, ToM and pain brain regions operate as specialized networks with synchronized responses, and (3) that maturity of the within-network and across-network correlations is related to ToM task performance in childhood.

In adults, each network exhibited strong positive correlations within-network, and strong negative correlations across network (within-ToM correlation $M$(s.e.) $= .48(.02)$; within-Pain correlation $M$(s.e.) $= .35(.02)$; across-network $M$(s.e.) $= -.17(.02)$; paired sample two-tailed $t$-tests ($n = 33$): within-ToM vs. across-network: $t(32) = 19.1$, $p < 2.2 \times 10^{-16}$; within-Pain vs. across-network: $t(32) = 23.2$, $p < 2.2 \times 10^{-16}$). See Methods,

Supplementary Fig. 1, and Supplementary Table 2 for details about the regions of interest.

This pattern of network correlations strengthened substantially between the ages of 3 and 12 years (Fig. 2; Supplementary Fig. 2 and 3). Among children, within-ToM and within-Pain network correlations increased significantly with age (Spearman partial correlation test, including motion (number of artifact timepoints) as a covariate ($n = 122$): within-ToM: $r_s(119) = .37$, $p < .00005$; within-Pain: $r_s(119) = .28$, $p = .002$). Across-network correlations decreased significantly with age (Spearman partial correlation test, including motion as a covariate ($n = 122$): $r_s(119) = -.35$, $p < .0001$). Within and across-network correlations were significantly greater in adults, compared to children (linear regression testing for effects of age group and motion on within-ToM correlation: effect of group (child ($n = 122$) vs. adult ($n = 33$)): $b = -.97$, $t = -5.7$, $p < 6.2 \times 10^{-8}$, effect of motion: $b = -.3$, $t = -4.3$, $p < .0001$; linear regression testing for effects of age group and motion on within-Pain correlation: effect of group (child ($n = 122$) vs. adult ($n = 33$)): $b = -.75$, $t = -3.8$, $p = .0002$, effect of motion: $b = -.03$, $t = -.31$, $p = .8$; linear regression testing for effects of age group and motion on across-network correlation: effect of group (child ($n = 122$) vs. adult ($n = 33$)): $b = 1.26$, $t = 7.2$, $p = 2.2 \times 10^{-11}$, effect of motion: $b = .07$, $t = .94$, $p = .4$). To ensure that developmental changes in correlation strength were not driven by various aspects of data quality (such as improved co-registration with age), we conducted inter-region correlation analyses on face and scene brain regions as well as bilateral primary motor and visual cortices; see Supplementary Fig. 3. These analyses showed that inter-region correlations in other networks (e.g., the face network and primary visual areas) do not show age-related change.

Nevertheless, the two networks were already functionally distinct in the youngest group of children we tested. In 3-year-old children only ($n = 17$), both ToM and pain networks had positive within-network correlations (within-ToM correlation $M$(s.e.) $= 21(.02)$; within-Pain correlation $M$(s.e.) $= .23(.02)$). Within-network correlations were higher than the across-network correlation (paired sample two-tailed $t$-tests ($n = 17$): within-ToM vs. across-network: $t(16) = 6.2$, $p < .00005$, within-Pain vs. across-network: $t(16) = 6.9$, $p < .00001$). By contrast, unlike adults, ToM and pain networks were not anti-correlated in 3 year olds (across-network correlation $M$(s.e.) $= .05(.02)$). However, significantly greater within- than across- network correlations suggests that ToM and pain networks are functionally distinct by age 3 years. The strongest within-network correlations in the 3 year olds were between homologous pairs

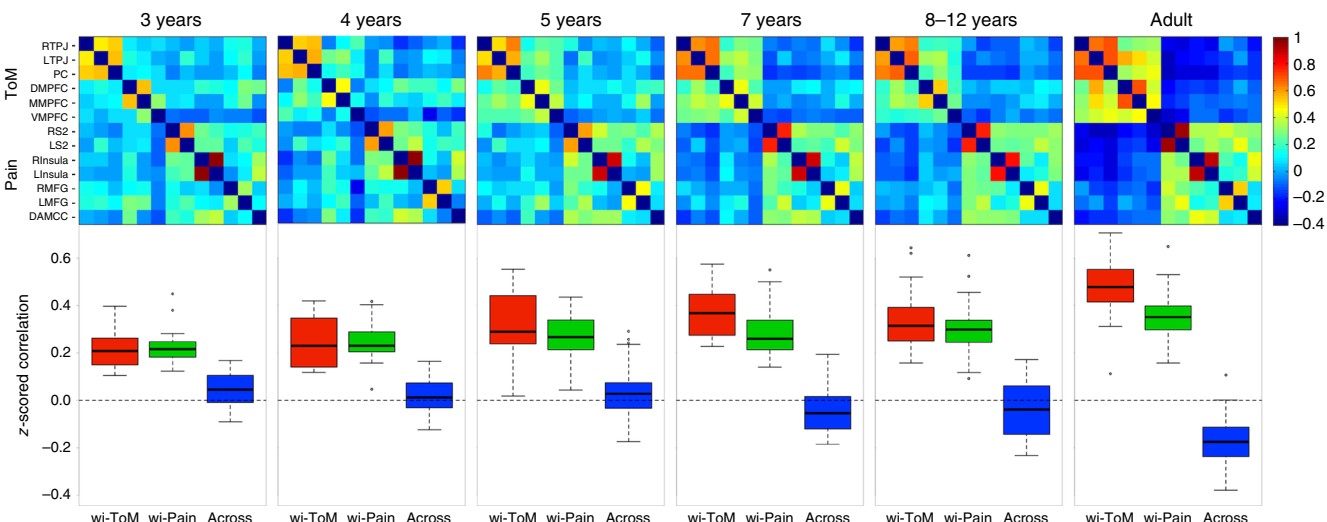

**Fig. 2** Inter-region correlation analysis. Top row: Average *z*-scored correlation matrices across all ToM and pain brain regions of interest (see *Y*-axis) per age group (3yo: *n* = 17; 4yo: *n* = 14; 5yo: *n* = 34; 7yo: *n* = 23; 8–12yo: *n* = 34; adults: *n* = 33). Regions are in the same order along the *X*-axes and *Y*-axes. Bottom row: boxplots of the within-ToM (red), within-Pain (green), and across-network (blue) *z*-scored correlation values per age group. Note that while data are binned into age groups here, age is a continuous variable in statistical tests

of regions in opposite hemispheres, such as right and left TPJ (ToM), and the right and left insula (Pain). These strong correlations, between pairs of regions that are functionally homologous but physically distant, suggest that even the data from 3 year old children are of high enough quality to detect inter-region correlations when they exist; and therefore that changes with age in other inter-region correlations reflect real changes in the functional relationships between those regions. However, the functional separation of the two networks was not fully explained by the strong correlations between bilateral pairs (Within-non-bilateral-ToM correlation *M*(s.e.) = .20(.02), Within-non-bilateral-Pain correlation *M*(s.e.) = .17(.02); paired sample two-tailed *t*-tests (*n* = 17): within-non-bilateral-ToM vs. across-network: *t*(16) = 5.1, *p* = .0001, within-non-bilateral-Pain vs. across-network: *t*(16) = 4.4, *p* = .0005).

In children, the strength of inter-region correlations within the ToM network was positively correlated with behavioral performance on the ToM battery outside the scanner (Kendall tau partial correlation test, including motion as a covariate (*n* = 122): *r*$_k$(119) = .23, *p* = .0002). The anti-correlation of ToM and pain networks was also correlated with ToM score (Kendall tau partial correlation test, including motion as a covariate (*n* = 122): *r*$_k$(119) = −.20, *p* = .001). However, there was no relationship between within-ToM or across-network correlations and ToM score when controlling for age in addition to motion (linear regressions testing for effect ToM score on within-ToM and across-network correlation, including age and motion as additional predictors (*n* = 122): NS effects of ToM score: ts < 1, *p* > .3).

We additionally tested for neural differences based on performance on explicit false-belief questions, among 3- to 5-year-old children. These questions were a subset of the questions in the ToM behavioral battery (see Methods). There was a significant difference in within-ToM network correlation between explicit false-belief task passers and failers (Within-ToM: Passers *M*(s.e.) = .29(.02), Failers *M*(s.e.) = .25(.03); linear regression testing for effects of FB-Group (pass vs. fail), age, and motion on within-ToM network correlation: effect of FB-Group (pass (*n* = 30) vs. fail (*n* = 15)): b = −.70, *t* = −2.06, *p* = .046, effect of age: b = .73, *t* = 4.4, *p* < .0005, effect of motion: b = −.34, *t* = −2.7, *p* = .009). This group difference becomes marginal when

response inhibition (DCCS summary score) is additionally included in the regression (effect of FB-Group (pass (*n* = 30) vs. fail (*n* = 15)): b = −.64, *t* = −1.80, *p* = .079, effect of age: b = .74, *t* = 4.4, *p* < .0001, effect of motion: b = −.33, *t* = −2.5, *p* = .02, NS effect of DCCS (response inhibition): b = −.08, *t* = −.59, *p* = .56). There was no difference in across-network correlation between these two groups (Passers *M*(s.e.) = .04(.02), Failers *M*(s.e.) = .03(.03); linear regression testing for effects of FB-Group (pass vs. fail), age, and motion on across-network correlation: NS effect of FB-group (pass (*n* = 30) vs. fail (*n* = 15)): b = .51, *t* = 1.2, *p* = .23, NS effect of age: b = −.29, *t* = −1.4, *p* = .16, NS effect of motion: b = −.004, *t* = −.02, *p* = .98). See Fig. 3a, b.

**Reverse correlation analysis.** Reverse correlation analyses are data-driven analyses used to identify events (>4 s) in a continuous naturalistic stimulus that evokes reliable positive hemodynamic responses in the same region across subjects[41]. Here we first use reverse correlation analyses to identify events that drive activity in ToM and pain brain regions, and subsequently test for developmental change in the magnitude of response to these events in children. As a first step, we successfully replicated previous results that responses in the fusiform gyrus are driven by face stimuli[41]; see Supplementary Fig. 4. Given these analyses have not yet been applied to pediatric data, this replication enabled us to be more confident in our analysis stream, the use of group regions of interest (ROIs), as opposed to individually defined ROIs, and the quality of our fMRI data (especially in young children, using a relatively short movie).

We applied reverse correlation analyses to the average response timecourses in the ToM network and pain matrix in adult participants. Because the inter-region correlation analysis suggested that ToM and pain regions comprise two functionally distinct networks by age three, we calculated the average timecourse across ROIs within each network. After identifying events based on the timecourse data from ToM and pain networks in adults, we extracted the response magnitude of each event from all child participants (see Methods). This analysis was used to determine (1) which events in the movie elicit the highest responses from ToM and pain regions in adults, (2) whether

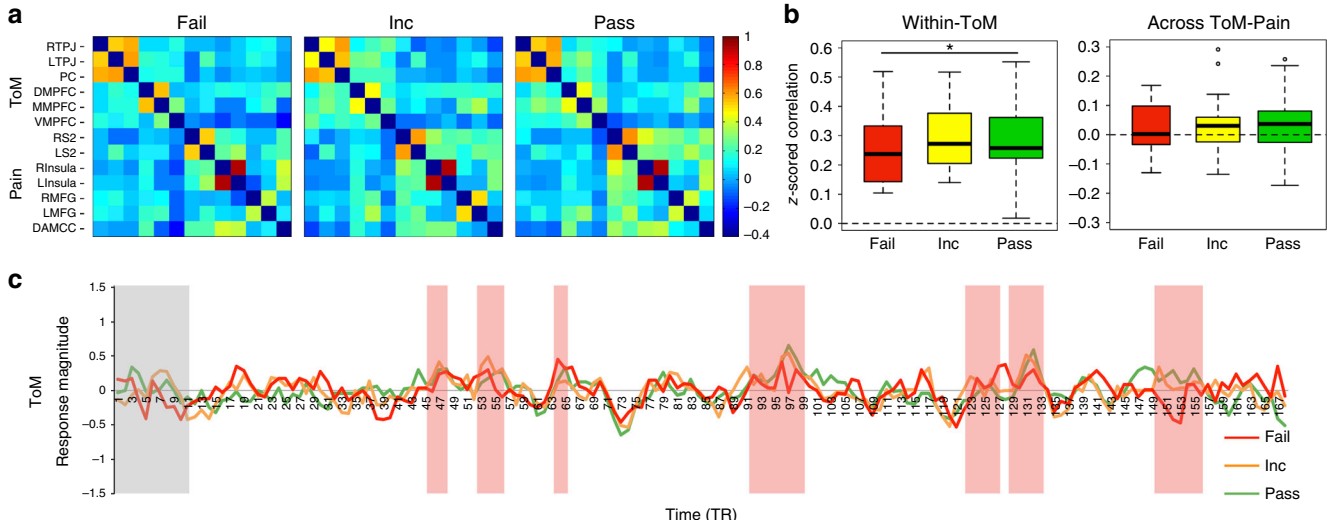

**Fig. 3** Similar functional responses in children who pass and fail explicit false-belief tasks. **a** Average z-scored correlation matrices for three to 5-year-old children who pass ($n = 30$), fail ($n = 15$), or perform inconsistently on ($n = 20$) explicit false-belief tasks. Regions are in the same order along the x-axes and y-axes. **b** Boxplots of z-scored correlation values within-ToM and across-ToM-Pain brain regions, based on false-belief task performance. Asterisk denotes a significant effect of false-belief task group (pass vs. fail) in a linear regression that also includes age and amount of motion (number of artifact timepoints) as covariates ($p < .05$); this group effect becomes marginal when additionally including a measure of response inhibition (DCCS). **c** Average timecourse of response in the ToM network for false-belief passers (green), failers (red), and inconsistent performers (orange), during viewing of 'Partly Cloudy'[61]

responses in ToM and pain regions in 3-year-old children are driven by the same events that drive corresponding responses in adults, and (3) the extent to which the responses to these events changes with age or ToM development in childhood.

In adults, the reverse correlation analysis produced seven theory of mind events (68 s total, $M$(s.d.) length 9.7(4.2) s) and twelve pain events (86 s total, $M$(s.d.) length 7.2(4.7)s); see Fig. 4a. All seven peak 'mind' events depict (changes in) the characters' beliefs, desires, and/or emotions: e.g., Gus is afraid that Peck will abandon him, Peck is embarrassed when Gus catches him gazing at another cloud. A majority of the 'body' events (8/12) depict the characters' physical pain (e.g., Peck being bitten by a crocodile) or transformations to the body (e.g. electricity changing a ball of cloud into a ram). The five events that have the highest response magnitude in each network in adults are shown in Fig. 4b; see Supplementary Fig. 5 for all events, Supplementary Table 3 for full descriptions of these events and timing and duration information, Supplementary Fig. 6 for a replication in an independent sample of adults, and Supplementary Fig. 7 for correspondence between these events and previously used hand-coded events. The timepoints that exceeded baseline for ToM and pain networks were almost entirely non-overlapping, with the exception of a single timepoint (2 s). This timepoint is the last timepoint of event T05, and the first timepoint of event P05; the response magnitude of both networks is significantly above baseline during this timepoint; see Fig. 4a. This extent of overlap is significantly less than that that would occur by chance (5/1000 random timecourse permutations with the same number and duration of ToM and Pain events have at most one timepoint of overlap; $p = .005$), and is present despite not regressing out a global signal from the timecourses of each network. See Supplementary Note 1 for a similar overlap analysis between face and ToM, and face and pain, events. These results converge with previous evidence for a similar functional division when participants read short verbal narratives, or when participants endogenously shift their attention to bodily versus mentalistic aspects of one movie or picture[2–5,38].

The average timecourse in ToM and pain regions in children was highly correlated with that of adults (pearson correlation tests between adult average timecourse and child average timecourse, TRs 11:168, for each child age bin: ToM: 3yo: $r = .28$, 4yo: $r = .31$, 5yo: $r = .60$, 7yo: $r = .72$, 8–12yo: $r = .82$ (all $p < .0005$; Bonferroni correction for multiple comparisons $\alpha = .01$, for five age bins); Pain: 3yo: $r = .60$, 4yo: $r = .56$, 5yo: $r = .73$, 7yo: $r = .83$, 8–12yo: $r = .89$ (all $p < 1.0 \times 10^{-13}$; $\alpha = .01$); see Supplementary Table 4). Nevertheless, we observed evidence of developmental change. Among children, three pain events (P01, P04, P08) and two ToM events (T01, T02) evoked significantly greater responses with age (spearman partial correlation tests, including motion as a covariate ($n = 122$); Pain: $p < .002$, $r_s s > .29$; ToM: $p < .0026$, $r_s s > .28$; Bonferroni correction for multiple comparisons $\alpha = .0026$, correcting for 19 events/tests). The two ToM events that showed greater responses with age are longer events that involve multiple and more complicated mental states (Supplementary Table 3). Responses in ToM regions during a third ToM event (T04) were significantly positively correlated with ToM score, controlling for age and motion (linear regression testing for effects of ToM score, age, and motion on T04 response magnitude ($n = 122$): effect of ToM score: $b = .4$, $t = 2.98$, $p = .0035$, NS effect of age: $b = -.14$, $t = -.99$, $p = .32$, NS effect of motion: $b = -.07$, $t = -.77$, $p = .45$; MC $\alpha = .007$, correcting for 7 ToM events/tests); see Fig. 1b. Response magnitude during ToM events did not differ significantly between children who pass and fail explicit false-belief tasks (all $p > .08$; linear regressions testing for effects of FB-Group (pass ($n = 30$) vs. fail ($n = 15$)), including age and motion as covariates); see Fig. 3c.

We next examined just the youngest children. As reported above, the overall timecourse of each network in 3 year olds ($n = 17$) was highly correlated with the average adult timecourses (pearson correlation test between adult average timecourse and average 3 year old timecourse, TRs 11:168: ToM: $r = .28$ $p = .00046$; Pain: $r = .60$, $p < 1.0 \times 10^{-15}$). Reverse correlation analysis conducted on the 3 year olds' data alone identified 4 of the 12 pain events and 1 of the 7 ToM events discovered in the

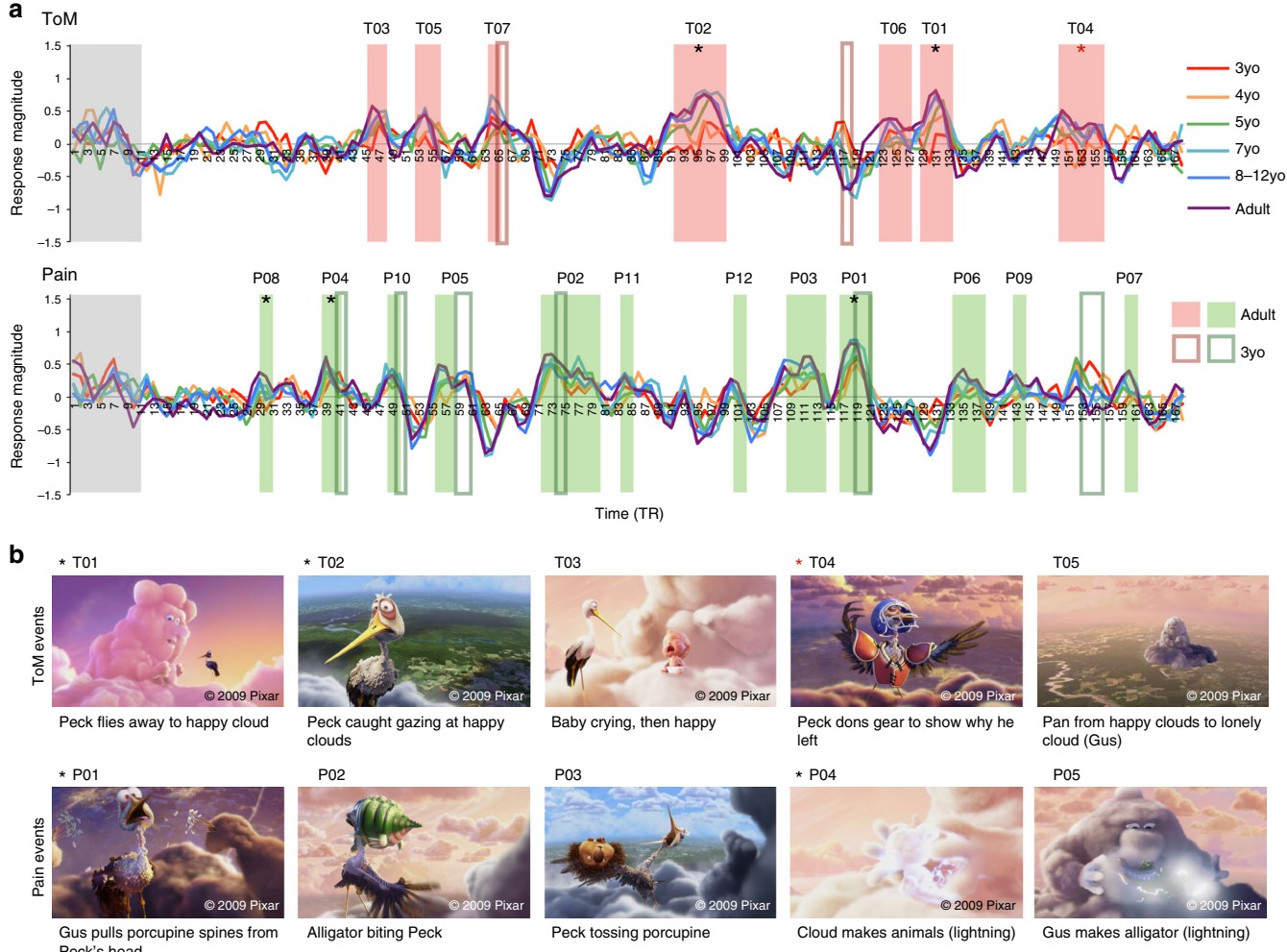

**Fig. 4** Reverse Correlation Analysis. **a** The average timecourse per age group for the ToM network (top) and Pain matrix (bottom), during viewing of 'Partly Cloudy'[61]. Each timepoint along the *x*-axis corresponds to a single TR (2 s); the entire movie was 168 TRs (<6 min). Shaded blocks show timepoints identified as ToM (red) and Pain (green) events in a reverse correlation analysis conducted on adults (*n* = 33); timepoints within the gray block correspond to the opening logos of the movie and were not analyzed. Dark red and green borders show timepoints identified as ToM and pain events, respectively, in 3-year-old children (*n* = 17). Event labels (e.g., T01, P01) indicate ranking of average magnitude of response in adults. Asterisks indicate significant positive correlations between peak magnitude of response and age (continuous variable; black) and ToM behavioral score (continuous variable; red), after correcting for multiple comparisons (age: 19 ToM/Pain events, $\alpha$ = .0026; ToM: 7 ToM events, $\alpha$ = .007). **b** Example frames and descriptions for the five events with the highest magnitude of response in adults, per network (see Supplementary Fig. 5 for all events, Supplementary Table 3 for fuller event descriptions and timing and duration information, and Supplementary Fig. 6 for a replication in an independent sample of adults). Images ©2009 Pixar, reused with permission. These images are not covered under the CC BY license for this article

adult sample. These events correspond to a subset of the timepoints that were identified as ToM or pain events in 3 year olds (Pain: 14/32 s, ToM: 4/8 s). Interestingly, 8 of the remaining 18 s identified as a pain event in 3-year-old children corresponds to a ToM event (T04) in adults, and the remaining 4 s identified as a ToM event corresponds to a pain event (P01) in adults (Fig. 4). The remaining 10 s identified as pain events occurred immediately after adult pain event timepoints.

**Relating functional maturity to inter-region correlations.** We tested whether the functional maturity (i.e., similarity to adults) of a child's movie-driven timecourse was related to the inter-region correlations measuring the child's network properties. Functional maturity was quantified by correlating each child's timecourse with the average adult timecourse. We found that the maturity of the movie-driven timecourse in both ToM and Pain networks was predicted by the anti-correlation of regions across networks (linear regressions testing for effects of across-network

correlation, within-network correlation, age, and motion on functional maturity measure ($n$ = 122): ToM: effect of across-network correlation: $b = -.4$, $t = -5.5$, $p = 2.2 \times 10^{-7}$, NS effect of within-ToM correlation: $b = .1$, $t = 1.5$, $p = .14$, effect of age: $b = .4$, $t = 5.3$, $p = 5.7 \times 10^{-7}$, NS effect of motion: $b = -.1$, $t = -1.5$, $p = .14$; Pain: effect of across-network correlation: $b = -.51$, $t = -7.4$, $p = 2.8 \times 10^{-11}$, NS effect of within-Pain correlation: $b = .13$, $t = 1.9$, $p = .06$, effect of age: $b = .3$, $t = 4.6$, $p = 1.3 \times 10^{-5}$, NS effect of motion: $b = -.08$, $t = -1.3$, $p = .2$); see Fig. 5.

That is, for children whose regions across the two networks showed more distinct responses, the average response within each network to the movie was more adult-like. This same pattern did not hold for within-network correlations. Greater within-network correlations were strongly associated with age, but not with timecourse maturity (linear regressions testing for effects of timecourse maturity, age, and motion on within-network correlations ($n$ = 122): ToM: NS effect of functional maturity:

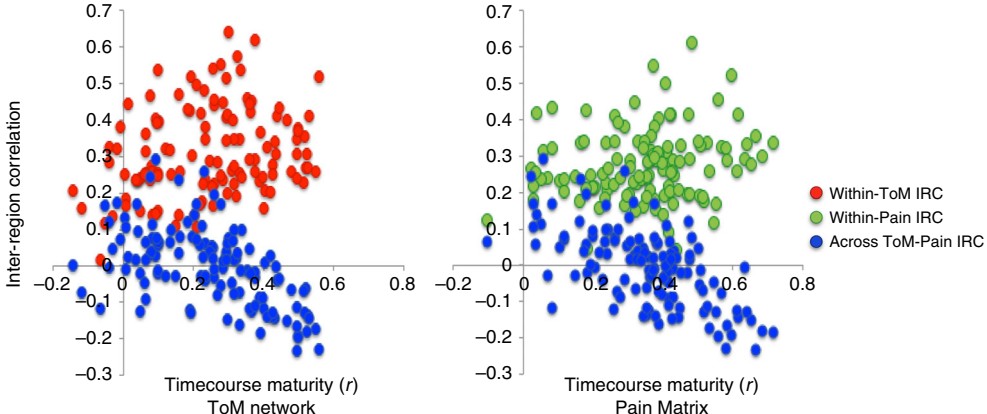

**Fig. 5** Relating functional maturity to inter-region correlations. In both networks, timecourse maturity (i.e., how correlated each child's timecourse is to the average adult timecourse (pearson's $r$), $x$-axis) is predicted by the extent to which the responses in ToM and Pain networks are anti-correlated ($z$-scored correlation values, $y$-axis). Both scatterplots show these values for all children ($n = 122$)

$b = .05$, $t = .48$, $p = .63$, effect of age: $b = .3$, $t = 2.5$, $p = .01$, effect of motion: $b = -.3$, $t = -3.56$, $p = .0005$; Pain: NS effect of functional maturity: $b = .07$, $t = .61$, $p = .55$, marginal effect of age: $b = .2$, $t = 1.96$, $p = .05$, NS effect of motion: $b = -.005$, $t = -.05$, $p = .96$). Additionally, while having an adult-like ToM timecourse was positively correlated with ToM behavior (spearman partial correlation test, including motion as a covariate ($n = 122$): $r_s(119) = .54$, $p = 1.3 \times 10^{-10}$), this relationship did not remain significant in a regression including age as an additional predictor (linear regression testing for effects of age, ToM score, and motion on functional maturity of ToM timecourse ($n = 122$): effect of Age: $b = .5$, t $= 4.2$, p $< .00005$, NS effect of ToM score: $b = .1$, t $= 1.2$, p $= .3$, effect of motion: $b = -.15$, t $= -2.1$, p $= .04$). Functional maturity of the ToM timecourse did not differ based on explicit false-belief task performance (linear regression testing for effects of FB-Group (pass vs. fail), age, and motion on functional maturity of ToM timecourse: NS effect of group (pass ($n = 30$) vs. fail ($n = 15$)): $b = -.12$, $t = -.35$, $p = .73$, effect of age: $b = .49$, $t = 2.9$, $p = .005$, effect of motion: $b = -.43$, $t = -3.4$, $p = .002$). Thus, among children, having functionally mature, task-driven responses is predicted by a child's anti-correlated responses in regions of the ToM and Pain networks.

## Discussion

Children's brains and their cognitive abilities undergo dramatic development in early childhood. In social cognition, for example, young children develop a remarkably sophisticated understanding of others' desires, thoughts and emotions, as distinct from their bodily reflexes, pains, and illnesses; much of this development occurs before children begin formal schooling at 6 years old[45–47]. Although brain regions involved in ToM have been extensively studied in adults, adolescents, and older children, fMRI experiments present serious obstacles for very young children. By using a short, engaging and naturalistic movie stimulus, we were able to collect functional data from a large sample of children ($n = 122$), including 65 children between 3 and 6 years of age. The movie stimulus, Pixar's 'Partly Cloudy,' depicts multiple events that focus on two aspects of the main characters (a cloud named Gus, and his stork friend Peck): their bodily sensations (often physical pain) and their mental states (beliefs, desires, and emotions). We measure developmental change in cortical networks recruited for reasoning about bodies (the pain matrix) and minds (the theory of mind network), and relate development in the ToM network to behavioral changes in theory of mind—bridging the gap between previous fMRI studies

in older children, and a large behavioral literature on early ToM development.

The first goal of this project was to measure developmental change in the pain matrix and theory of mind network. A key result emerged from multiple different analysis approaches: a core aspect of development in the social brain is the differentiation of spontaneous cortical responses to depictions of others' bodies versus minds. First, anti-correlations between the ToM and pain networks showed particularly dramatic change with age: regions in these two networks were uncorrelated in 3 year olds, but robustly anti-correlated in older children and adults. This anti-correlation predicted the maturity (i.e., similarity to adults) of each network's timecourse of response evoked by the movie. Second, while activity in ToM and pain networks in adults is driven by non-overlapping mentalistic and bodily events, respectively, in 3 year olds some events led to increased activity in the opposite network: the adult pain event P01 elicited activity in the ToM network, and the adult ToM event T04 elicited activity in the pain network of 3-year-old children. These results are in line with previous evidence that functionally selective brain regions respond less to non-preferred categories with age,[20,21,48,49] and suggest that development of functionally specialized brain regions for reasoning about others' internal states involves increasingly accurate application of specific neural resources (i.e., distinct groups of brain regions) to specific inputs (events depicting others' mental states versus physical sensations).

Almost all previous publications of timecourse data in young children describe analyses of resting state data: fMRI data collected while participants are not performing any particular cognitive task, or in some cases, while participants are asleep[50]. One advantage of measuring inter-region correlations during a movie, as we did here, is that children's psychological state (e.g., attention, anxiety, alertness) is likely more similar, across ages. On the other hand, a disadvantage is that we cannot distinguish between intrinsic and task-driven contributions to the inter-region correlations[51]. For example, the development of anti-correlations between ToM and pain networks may reflect a combination of both intrinsic changes in network structure, and increasing functional selectivity of the movie-driven response in individual regions[52]. Future studies could tease apart contributions of intrinsic and task-driven connectivity by collecting both resting-state and functional task data from the same child; however, for 3-year-old children any additional data collection within a session would be challenging.

The second goal of this project was to ask how change in the ToM network relates to children's theory of mind cognitive

abilities. All children were asked questions about other people's actions, beliefs, desires, expectations, and moral blameworthiness. Within this set of questions, six questions focused specifically on predicting and explaining actions based on false beliefs. The transition from failure to success on the false-belief task has sometimes been interpreted as evidence of discontinuity in development around age 4 years: the emergence of a new theory, or cognitive mechanism, that did not exist earlier[12–14]. A second possibility is that changes in executive function (e.g., response inhibition) unmask children's previously existing ToM[31–33]. A third possibility is that children's theory of mind itself undergoes continuous and gradual development, from relatively simple concepts of perceptions and goals in 2 year olds to a sophisticated understanding of negligence and irony in early adolescence[34–37,53]. Each of these possibilities makes different predictions for the patterns of neural data we measured here. Unlike any previous fMRI study of ToM, our sample included a substantial number of children who systematically failed explicit false-belief tests. This enabled us to test for signatures of neural responses that predict improved performance on false-beliefs tasks, in addition to ToM reasoning more generally.

Our data were most inconsistent with the prediction of a robust discontinuity in response, associated with the transition from failure to success on explicit false-belief tasks. In the profiles of neural responses, we saw no major discontinuity when children begin to systematically pass false-belief tasks. Brain regions involved in ToM in adulthood already constitute a distinct network in 3-year-old children, which gradually becomes more integrated and distinct from other networks over the next decade. Similarly, the timecourse of response in the ToM network in response to a social movie is strongly positively correlated, even between 3 year olds and adults. The timecourse and peak event responses show gradual continuous development over childhood. Focusing specifically on 3- to 5-year-old children, the neural responses to social movies in children who systematically fail versus pass explicit false-belief tasks were similar: there were no differences in the magnitude of response to the seven ToM events identified using reverse correlation analyses, and no difference in the extent of anti-correlation of the responses in ToM and pain networks. Consistent with recent evidence that false-belief passers have increased structural connectivity between ToM brain regions, compared to failers[54], we find that passing false-belief tasks was associated with increased functional correlations among regions in the ToM network, but this group difference became marginal when taking response inhibition abilities into account, and the same neural measure was also associated with age in the full sample.

Our data were partially consistent with the prediction that spontaneous processing of others' mental states within domain-specific regions for ToM is similar, regardless of performance on explicit false-belief tasks. Research in adults suggests that the same ToM brain regions are recruited to reason about mental state content, regardless of whether the stimulus is verbal or nonverbal, instructed or spontaneous[19,38,55,56]. Spontaneously generating mentalistic descriptions of actions is a precursor of performance on explicit tasks[57], and is correlated with cortical thinning of ToM regions in adults[58]. In the current study, 3-year-old children who systematically fail false-belief tasks nevertheless recruited ToM brain regions at similar times in the movie and as a distinct network from the pain matrix. On the other hand, we did observe significant change within ToM brain regions, and in the dissociation between ToM and pain networks, which is not predicted by the view that explicit ToM tasks measure change in domain general performance limitations.

Overall, our results seem most consistent with the prediction that a distinct neural response to others' minds versus bodies is

already beginning to develop well before children explicitly pass false-belief tasks, and continues to develop well after[7,8,47]. For example, for one event in the movie, the magnitude of response in the ToM network correlated with the child's score on the full ToM battery (not limited to false belief items). This event (T04) shows Peck donning protective football gear in front of Gus. In context, this event depicts Gus revising previous beliefs and emotions (because Gus believed that Peck had abandoned him, Gus had been furious and devastated; once Peck shows Gus the helmet and pads, Gus realizes that Peck has not abandoned him, and indeed never intended to abandon him, and Gus feels happy and relieved). Increased activity in ToM regions during this event may reflect children's improved ability to consider the relevance of the current event for (past) beliefs or emotions that are not explicitly depicted[59].

These fMRI results are thus consistent with evidence in developmental psychology for slow, continuous development of theory of mind. In individual children, the transition from failing to passing explicit false-belief tasks occurs gradually and noisily: children who begin to answer explicit false-belief questions correctly often subsequently fall back to incorrect responses[57]. Improvement is boosted by explicit explanatory practice and feedback over a relatively long period of time. The noisiness of development is visible in the current dataset: twenty children answered three or four out of six explicit false-belief questions correctly, within a single testing session. Also, mastering explicit false-belief tasks is not equivalent to having a fully mature theory of mind[60]; older children are still learning to infer hidden emotions[34], discriminate degrees of moral blameworthiness[53], and understand non-literal speech like sarcasm and irony[37]. On this view false-belief task performance is likely just one step along a long trajectory of increasingly sophisticated understanding of other minds.

In sum, we report evidence that when people spontaneously watch an animated movie evoking the internal states of others, distinct networks of cortical regions are recruited for events that make salient internal states of the mind versus of the body. These networks are already functionally distinct in 3-year-old children, but show increasing within-network and decreasing across-network correlations throughout childhood. The anti-correlation of the two networks strongly predicts the maturity of each network, in response to the movie. Specific peak events within the movie evoke activity that increases with age, and with theory of mind reasoning ability. On the other hand, the most famous milestone in ToM behavioral development, passing explicit false-belief tasks, does not correspond with a discontinuity in the neural basis for reasoning about the minds of others.

## Methods

**Participants**. One hundred twenty two 3.5–12-year-old children ($M$(s.d.) = 6.7 (2.3); 64 females) participated in the study. 110 children were right-handed and 3 were ambidextrous (as indicated by parent or legal guardian). This sample includes 65 children under the age of 6 years ($M$(s.d.) = 4.82(.81) years; 34 females; 54 RH/3 Ambi); this subset of children were used to test for neural differences between children who pass ($n = 30$; $M$(s.d.) = 5.2(.70); 15 females; 26 RH/2 Ambi) and fail ($n = 15$; $M$(s.d.) = 4.08(.56); 6 females; 11 RH/4 Ambi) false-belief tasks. Twenty children in this subset responded inconsistently to false-belief tasks ($M$(s.d.) = 4.75 (.73); 13 female; 17 RH/1 Ambi). An additional 19 children were recruited to participate and excluded from all analyses for not completing or participating in the study ($n = 12$), language delays ($n = 2$), and excessive motion during the fMRI scan ($n = 5$; see fMRI Data Analysis for details). Thirty three adult participants (ages 18–39 years; $M$(s.d.) = 24.8(5.3); 20 females; 32 RH/1 LH) additionally participated in the fMRI portion of the study. Child and adult participants were recruited from the local community. All adult participants gave written consent; parent/guardian consent and child assent was received for all child participants. Recruitment and experiment protocols were approved by the Committee on the Use of Humans as Experimental Subjects (COUHES) at the Massachusetts Institute of Technology.

**fMRI stimuli**. Participants watched a silent version of 'Partly Cloudy,'[61] a 5.6-min animated movie[38]. A short description of the plot can be found online (https://www.pixar.com/partly-cloudy#partly-cloudy-1). Previous research suggests that pediatric populations move significantly less during fMRI scans using movie stimuli[62]. The stimulus was preceded by 10 s of rest, and participants were instructed to watch the movie and remain still. Participants aged five and older completed additional tasks prior to viewing this stimulus; these tasks largely involved listening to (children) or reading (adults) stories.

**fMRI data acquisition**. Prior to the scan, child participants completed a mock scan in order to become acclimated to the scanner environment and sounds, and to learn how to stay still. Children were given the option to hold a large stuffed animal during the fMRI scan in order to feel calm and to prevent fidgeting. An experimenter stood by child participants' feet, near the entrance of the MRI bore, to ensure that the participant remained awake and attentive to the movie. If this experimenter noticed participant movement, she placed her hand gently on the participant's leg, as a reminder to stay still.

Whole-brain structural and functional MRI data were acquired on a 3-Tesla Siemens Tim Trio scanner located at the Athinoula A. Martinos Imaging Center at MIT. Children under age 5 years used one of two custom 32-channel phased-array head coils made for younger ($n = 3$, $M$(s.d.) = 3.91(.42) years) or older ($n = 28$, $M$(s.d.) = 4.07(.42) years) children[63]; all other participants used the standard Siemens 32-channel head coil. T1-weighted structural images were collected in 176 interleaved sagittal slices with 1 mm isotropic voxels (GRAPPA parallel imaging, acceleration factor of 3; adult coil: FOV: 256 mm; kid coils: FOV: 192 mm). Functional data were collected with a gradient-echo EPI sequence sensitive to Blood Oxygen Level Dependent (BOLD) contrast in 32 interleaved near-axial slices aligned with the anterior/posterior commissure, and covering the whole brain (EPI factor: 64; TR: 2 s, TE: 30 ms, flip angle: 90°). As participants were initially recruited for different studies, there are small differences in voxel size and slice gaps across participants (3.13 mm isotropic with no slice gap ($n = 5$ adults, $n = 3$ 7yos, $n = 20$ 8–12yo); 3.13 mm isotropic with 10% slice gap ($n = 28$ adults), 3 mm isotropic with 20% slice gap ($n = 1$ 3yo, $n = 3$ 4yo, $n = 2$ 7yo, $n = 1$ 9yo); 3 mm isotropic with 10% slice gap (all remaining participants)); all functional data were subsequently upsampled in normalized space to 2 mm isotropic voxels. Prospective acquisition correction was used to adjust the positions of the gradients based on the participant's head motion one TR back[64]. 168 volumes were acquired in each run; children under age five completed two functional runs, while older participants completed only one run. For consistency across participants, only the first run of data was analyzed. Four dummy scans were collected to allow for steady-state magnetization.

**fMRI data analysis**. FMRI data were analyzed using SPM8 (http://www.fil.ion.ucl.ac.uk/spm)[65] and custom software written in Matlab and R. Functional images were registered to the first image of the run; that image was registered to each participant's anatomical image, and each participant's anatomical image was normalized to the Montreal Neurological Institute (MNI) template. This enabled us to use group regions of interest (ROIs) and hypothesis spaces created in adult data sets, and to directly compare responses between child and adult participants. Previous research has suggested that anatomical differences between children as young as 7 years are small relative to the resolution of fMRI data, which supports usage of a common space between adults and children of this age (for similar procedures with children under age seven, see refs [21,66,67]; for methodological considerations, see ref. [68]). Registration of each individual's brain to the MNI template was visually inspected, including checking the match of the cortical envelope and internal features like the AC–PC and major sulci. All data were smoothed using a Gaussian filter (5 mm kernel).

Artifact timepoints were identified via the ART toolbox (https://www.nitrc.org/projects/artifact_detect/)[69] as timepoints for which there was (1) >2 mm composite motion relative to the previous timepoint or (2) a fluctuation in global signal that exceeded a threshold of three s.d. from the mean global signal. Participants were dropped if one-third or more of the timepoints collected were identified as artifact timepoints; this resulted in dropping five child participants from the sample (see Participants). Number of artifact timepoints differed significantly between child and adult participants (Child ($n = 122$): $M$(s.d.) = 10.5(10.6), Adult ($n = 33$): $M$(s.d.) = 2.8(4), Welch two-sample $t$-test: $t(137.7) = 6.49$, $p < .000001$). Among children, number of motion artifact timepoints was not correlated with age (spearman correlation test ($n = 122$): $r_s(120) = .02$, $p = .86$) or ToM score (kendall tau correlation test ($n = 122$): $r_k(120) = −.005$, $p = .94$). Number of artifact timepoints did not differ between young (3–5-year old) children based on false-belief task performance (linear regression tests for effect of FB-Group on number of motion artifact timepoints: NS effect of FB-group (Pass ($n = 30$) vs. Fail ($n = 15$)): $b = −.04$, $t = −.12$, $p = .9$; NS effect of FB-group (Pass ($n = 30$), Inc ($n = 20$), or Fail ($n = 15$)): $b < .05$, $p > .9$) or response inhibition (linear regression test for effect of DCCS on number of motion artifact timepoints ($n = 64$): NS effect of DCCS summary score: $b = .16$, $t = 1.18$, $p = .25$). See Supplementary Fig. 8 for visualization of the amount of motion per age group. Despite amount of motion being matched across children, and therefore likely not driving developmental effects within the child sample, we include number of motion artifact timepoints as a covariate in all analyses. Number of artifact timepoints is highly correlated with

measures of mean translation, rotation, and distance ($r > .8$). Because this measure is not normally distributed, spearman correlations were used when including amount of motion as a covariate in partial correlations.

Region of interest (ROI) analyses were conducted using group ROIs. ToM and pain matrix group ROIs were created in an independent group of adults ($n = 20$), scanned by Evelina Fedorenko and colleagues. These data were preprocessed and analyzed with procedures identical to those used for participants in the current study. Reverse correlation analyses were conducted in this separate group of adults, using 10 mm group ROIs surrounding peaks reported in previous publications (ToM regions[70]; Pain matrix[71]). Seven ToM and nine pain events were identified (ToM: 60 s total, $M$(s.d.) length: 8.6(4.6)s, Pain: 66 s total, $M$(s.d.) length: 7.3(4.4)s). We subsequently used a general-linear model to analyze BOLD activity of these participants as a function of condition, using these events. Second-level random effects analyses were used to examine the group-level response to Mental > Pain and Pain > Mental ($p < .001$, k = 10, uncorrected). We then drew 9 mm spheres surrounding the peak activation in each region, to create new group ROIs that were tailored to the stimulus, but defined in an independent sample of adults (see Supplementary Fig. 1 and Supplementary Table 2 for more information on all group ROIs, and Supplementary Fig. 7 for details of the convergence between events across the two adult samples and ROIs).

All timecourse analyses were conducted by extracting the preprocessed timecourse from each voxel per group ROI. We applied nearest neighbor interpolation over artifact timepoints (for methodological considerations on interpolating over artifacts before applying temporal filters, see refs [72,73]), and regressed out two kinds of nuisance covariates to reduce the influence of motion artifacts: (1) motion artifact timepoints; and (2) five principle component analysis (PCA)-based noise regressors generated using CompCor within individual subject white matter masks[74]. White matter masks were eroded by two voxels in each direction, in order to avoid partial voluming with cortex. CompCor regressors were defined using scrubbed data (e.g., artifact timepoints were identified and interpolated over prior to running CompCor).

For inter-region correlation analyses only, we additionally regressed out the raw timecourse extracted from bilateral primary motor cortex (M1). Primary motor cortex ROIs were 10 mm spheres drawn around peak coordinates generated with Neurosynth (http://neurosynth.org/; search term: "primary motor," forward inference from 273 studies; coordinates: [38,−24,58], [−38,−20,58]). These ROIs are included in the expanded inter-region correlation analysis shown in Supplementary Fig. 4; the bilateral M1 timecourse was not regressed out for this supplemental analysis. However, because this analysis showed that the within-M1 inter-region correlation increases with age among children, we regressed the bilateral M1 timecourse from the ToM and Pain timecourses for the inter-region correlation analyses reported in the main text, to ensure that the age effects in the ToM and pain networks are above and beyond developmental effects present in regions like primary motor cortex, and that within-network correlations are not falsely inflated by commonalities in signal fluctuation across the brain.

The residual timecourses were then high-pass filtered with a cutoff of 100 s. Timecourses from all voxels within an ROI were averaged, creating one timecourse per group ROI, and artifact timepoints were subsequently excluded (NaNed).

In inter-region correlation analyses, each ROI timecourse was correlated with every other ROI's timecourse, per subject, and these correlation values were Fisher $z$-transformed. Within-ToM correlations were the average correlation from each ToM ROI to every other ToM ROI, within-Pain correlations were the average correlation from each Pain ROI to every other Pain ROI, and across-network correlations were the average correlation from each ToM ROI to each Pain ROI. This procedure is similar to that used by ref. [42]. In order to test for developmental change in within-network and across-network correlations, we conducted linear regressions to test for (1) significant differences between adults and children, in regressions that included group (child vs. adult) and number of artifact timepoints as predictors, (2) significant effects of age (as a continuous variable), ToM performance, and number of artifact timepoints among children, and (3) significant group differences between children who pass and fail explicit false-belief tasks, including number of artifact timepoints and age as predictors. In order to test whether ToM and pain networks are coherent and specialized early in childhood, we used t-tests to compare within-network versus across-network correlations in 3-year-old children ($n = 17$). Within-network and across-network correlation measures were normally distributed ($p > .22$, one-sample Kolmogorov–Smirnov tests), and variance in within-ToM, within-Pain, and across-network correlations did not differ across children and adults, or false-belief passers vs. failers ($F$-tests to compare two variances: children ($n = 122$) vs. adults ($n = 33$): $F(32,121) > 1.1$, $p > .66$; pass ($n = 30$) vs. fail ($n = 15$): $F(14,29) > .78$, $p > .65$).

Initial reverse correlation analyses were conducted on adult participants only. Each ROI timecourse was $z$-normalized, and timecourses within each network were averaged across ROIs, resulting in one timecourse for face regions, ToM regions, and the pain matrix per adult participant. Except for the first 10 timepoints (5 TRs rest, followed by 5 TRs of the movie introduction (Disney castle and Pixar logos)), the residual signal values across adult subjects for each timepoint were tested against baseline (0) using a one-tailed $t$-test. This procedure is similar to that used by[41]. Events were defined as two or more consecutive significantly positive timepoints within each network. Events were rank-ordered according to the average magnitude of response to the peak timepoint in adults, and labeled

according to the ordering (e.g., event T01 is the ToM event that evoked the highest magnitude of response in the ToM network).

In adults, we conducted an overlap analysis to determine whether the number of timepoints labeled as both ToM and pain events was statistically fewer than would occur by chance. We constructed 1000 permutations of ToM and pain timecourses, which had the same number and duration of events. The constructed timecourses were 158 TRs in length (the experiment was 168 TRs; the first 10 TRs were excluded from the reverse correlation analysis because the movie started on TR 11). For each permutation, we randomly scrambled the order of ToM and pain events. We then filled in the timepoints between events with zeros, with a random proportion of zeros between events such that the total number of zeros was equal to the total number of non-event timepoints in the original timecourses (ToM: 125 TRs; Pain: 116 TRs). Events within a timecourse (ToM or Pain) necessarily had to be separated by at least one timepoint, since they would otherwise be counted as a single event. The first event of each timecourse could be preceded by zero zeros, and the last event of each timecourse could be followed by zero zeros. We calculated the sum of the number of timepoints tagged as ToM and pain events in each pair of permutations (ToM and pain timecourses), and subsequently calculated the proportion of permutations that resulted in the same or a smaller amount of overlap as observed in the reverse correlation analysis.

In order to test for developmental effects in the magnitude of response to ToM and pain events, we defined a peak timepoint per event as the timepoint with the highest average signal value in adults, and tested for significant correlations between the magnitude of response at peak timepoints and age (as a continuous variable), including amount of motion (number of artifact timepoints) as a covariate. Because this measure of motion is non-normally distributed, we employed spearman correlations. For ToM regions only, we used linear regressions to test for a significant relationship between peak magnitude of response and theory of mind behavior (overall, in all children), and to test whether responses at peak timepoints differed between children who pass ($n = 30$) and fail ($n = 15$) explicit false-belief tasks. Response magnitude at all peak events was normally distributed (all $p > .23$, one-sample Kolmogorov–Smirnov test). Response magnitudes showed similar variance across false-belief task passers ($n = 30$) and failers ($n = 15$) ($F$-tests to compare two variances: all $F(13, 28) > .7$, $p > .07$), with the exception of one event (T03: $F(14, 28) = .30$, $p = .02$). A permutation test was used to test for group differences in magnitude of response to this event[75]. We ran the reverse correlation analysis in 3-year-old participants only ($n = 17$), in order to examine response specificity at this young age, and to better understand developmental differences.

Finally, we tested whether the functional maturity of each child's timecourse responses (i.e., similarity to adults) was related to the inter-region network correlations. We calculated the pearson correlation between each child's ToM timecourse (averaged across ROIs) and the average adult ToM timecourse; we similarly calculated the pearson correlation between each child's pain matrix timecourse and the average adult pain matrix timecourse. The timecourses used for this analysis were the same as those used for the reverse correlation analysis, prior to $z$-normalization (TRs 11:168). We tested whether, across children, this measure of functional maturity per network was correlated with within-network and across-network inter-region correlations, or related to ToM behavior. The neural maturity measure was normally distributed in both networks ($p > .29$, one-sample Kolmogorov–Smirnov test). Variance in this measure in the ToM network did not differ between children who pass ($n = 30$) and fail ($n = 15$) false-belief tasks ($F$ test to compare two variances: $F(14, 29) > 1.00$, $p > .95$). We additionally calculated and report the pearson correlation between the average timecourse of children in each age group and the average adult timecourse.

All of the analyses reported in this manuscript should be considered exploratory, not confirmatory, in that the analyses described here were not chosen prior to data collection, and data collection was not completed with this specific set of analyses in mind. While we deliberately chose this stimulus in order to measure neural responses in very young children (ages 3–4 years), older children visited the lab to participate in a different study, and additionally completed the protocol of the current study. We then recognized the opportunity of analyzing the full cross-sectional dataset, and chose analyses based on the stimulus (time series analyses seemed to utilize more data and be more sensitive than previous analysis methods[38]), and on recent relevant progress in the field[42,76,77].

**Behavioral battery**. After the scan, all children completed a behavioral task battery including (in order) an explicit theory of mind battery and a measure of nonverbal IQ (under 5 years: WPPSI block design[78], over 5 years: nonverbal KBIT-II[79]). Children under age seven then completed a computerized version of the Dimensional Change Card Sort task as a measuring of response inhibition. Performance on DCCS was captured using the summary score[44]; one child (an inconsistent FB task performer) failed to complete the DCCS task.

**Explicit ToM task and false-belief composite score**. All children completed a custom-made explicit ToM battery[21] (https://osf.io/G5ZPV/), which involved listening to an experimenter tell a story and answering prediction and explanation questions that required reasoning about the mental states of the characters. Because this task was designed to capture variability in ToM reasoning across a wide age-range of children, the questions varied in difficulty. Easier items involved reasoning

about similar and diverse desires, true beliefs, and emotion prediction; harder items included reasoning about false beliefs, moral blame-worthiness, and second-order false beliefs. Two analogous booklets were used; children ages 3–4 and 10–12 years old listened to a story about students finding snacks, and 5-year-old children listened to a story about students finding books; 7–9 year-old-children were split among the books (snacks: $n = 16$; books: $n = 33$). Different booklets were used across children because children of different ages participated in different studies that all involved the current protocol. However, the two booklets were designed for repeated measures designs: analogous stories and questions across the two booklets had identical syntax, but different semantic content: one story was about helping children find their books, the other was about finding snacks. A previous study using the 'finding books' booklet suggests the validity of this task to capture theory of mind development in children ages 5–12 years old[21]. These booklet tasks and instructions are available on the Open Science Framework (https://osf.io/G5ZPV; DOI: 10.17605/OSF.IO/G5ZPV; ARK: c7605/osf.io/g5zpv).

Each child's performance on the ToM battery was summarized as the proportion of questions answered correctly, out of 24 matched items (14 prediction items and 10 explanation items). An additional two control items were asked to ensure that children were paying attention; after ensuring all children answered these questions correctly, these items were not further analyzed. Children ages 3–5 years old were also categorized based on their performance on a false-belief composite score based on six explicit false-belief questions (4 prediction, 2 explanation) within the ToM booklet. These six questions were chosen because they were canonical explicit false-belief questions describing changes in location or unexpected contents[11,12,80]. The composite score demonstrated acceptable reliability (Cronbach's $\alpha = .71$). Children were categorized as explicit false-belief 'passers' if they answered five or six out of six false-belief questions correct, 'inconsistent performers' if they answered three or four questions correct, and 'failers' if they answered zero to two questions correct.

We tested for significant correlations between age, DCCS and ToM, and for differences in these scores between children who pass and fail false-belief tasks. We used Kendall's rank correlation tau, given non-normal distributions of the ToM score (Shapiro–Wilk normality test: $w = .9$, $p < .00001$) and DCCS score ($w = .75$, $p < .00001$), and given the frequency of ties in both of these measures.

**Code availability**. The analysis code used to generate the findings of this study are available from the corresponding author upon request.

**Data availability**. The fMRI and behavioral data collected and analyzed during the current study are available through the OpenfMRI project (https://openfmri.org/; Link: https://www.openfmri.org/dataset/ds000228/ DOI: 10.5072/FK2V69GD88). The ToM behavioral battery is additionally available through OSF (https://osf.io/G5ZPV/; DOI: 10.17605/OSF.IO/G5ZPV; ARK: c7605/osf.io/g5zpv). The corresponding author welcomes any additional requests for materials or data.

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

## Acknowledgements

We thank the Athinoula A. Martinos Imaging Center at the McGovern Institute for Brain Research at MIT, Jorie Koster-Hale, Natalia Velez-Alicea, Mika Asaba, and Nir Jacoby for help with data collection, and Stefano Anzellotti, Dorit Kliemann, Julia Leonard, and Lindsey Powell for helpful feedback and discussion. We thank Hyowon Gweon for development of the theory of mind behavioral battery, and Todd Thompson for helping to make the data available. We thank members of the Fedorenko lab for providing the data for the replication experiment. In particular, Alex Paunov and Zach Mineroff led the data collection effort, with help from Caitlyn Hoeflin, Amaya Arcelus, Brianna Pritchett, Idan Blank, and Cara Borelli. We also gratefully acknowledge support of this project by a NSF Graduate Research Fellowship (#1122374 to H.R.), and an NSF CAREER award (#095518 to R.S.), NIH R01-MH096914-05, a Middleton Chair grant (R.S.), and support from the David and Lucile Packard Foundation (#2008-333024 to R.S.).

## Author contributions

H.R. and R.S. devised the experiment. G.L. recruited all participants. H.R. G.L. and A.R.-N. collected the data, all authors analyzed the data, and H.R. and R.S. wrote the manuscript.

## Additional information

**Competing interests:** The authors declare no competing interests.

