## [Peer Review File · Nature Communications]

Reviewers' comments:

Reviewer #1 (Remarks to the Author):

This study investigates the developmental trajectories of two functional brain networks, Theory of Mind (ToM) and Pain, which have been previously defined and studied in adults. A large cohort of children (122) ages 3-12, as well as a group of adults (33), watched a 5.6 minute animated movie during fMRI. Subjects also underwent behavioral testing for ToM. In adults, ToM and Pain networks are 1) strongly correlated within-network and 2) negatively correlated across network, as previously observed in the literature. A main result of the current paper is that both effects strengthen gradually as age increases. In children, overall within-network ToM correlation was positively related to behavioral performance on the ToM behavioral battery, as was anti-correlation of ToM and pain networks. This effect disappears when controlling for age. When restricting the battery to questions of explicit false belief, among 3-5-yo children there was a significant difference in within-ToM network correlation between passers and failers. The maturity of the movie-driven timecourse (as measured by the similarity to the adult neural timecourse) in both ToM and Pain networks was predicted by the anti-correlation of regions across networks. Greater within-network correlations were strongly associated with age, but not with timecourse maturity, for ToM and Pain networks. ToM timecourse maturity was not significantly correlated with ToM behavior.

The questions and analysis approach of the study are very interesting, and the dataset is impressive. The manuscript is well-written and the analyses are clearly described. I believe the results will be of interest to many researchers, and the public release of the dataset would benefit the community. I do have concerns about a few of the analyses, as well as clarification questions.

1. In adults, ToM and Pain networks are 1) strongly correlated within-network and 2) negatively correlated across network, as previously observed in the literature. A main result of the current paper is that both effects strengthen as age increases. Network ROIs are defined from an independent adult group and then placed onto the MNI-aligned child brain. Is it possible that the above result (effects strengthening with age) is due to increasingly better match (coregistration) between the child brain and adult brain with increasing child age? To illustrate: if you had two adult datasets (Groups 1 and 2), and simply added noise to the coregistration parameters of one of these datasets (Group 2) such that the individuals in Group 2 were less well aligned with a) Group 1 and b) each other within the group, and performed the same analysis procedures, would you see decreased within-network correlation and less negative between-network correlation? If you added more noise, would these effects reduce further, i.e., the same overall pattern of results as in the current study?

I understand that there is some nuance to this question; children's brains are surely becoming anatomically more like adult brains with age, and this problem is deeply entangled with the goal of understanding how children's brains are becoming functionally more like adult brains. However, I think this is a legitimate possible confound that should be addressed.

Some ideas for approaching the problem:

- Run a simulation with a separate adult dataset as I described above, adding varying amounts of noise to coregistration parameters (or some equivalent manipulation of noise).
- Present a control network that does not show the pattern of increasing within-network correlation with increasing age. Note that in order to be a good comparison, this control network should be comparable to ToM and Pain in its adult within-network correlation.
- Put a functional connectivity seed in PCC for each child brain and calculate how much the map overlaps (Dice or Jaccard index) with the same PCC-seed maps in the adult brain and with other subjects of the same age. This can be plotted out for different correlation thresholds, and could help give a sense for whether network variability is substantially higher in the child brain in general.

In short, some demonstration that the results do not stem entirely from decreasing child-adult coregistration error with child age seems necessary in order to support the claim that ToM network responses "show gradual continuous development over childhood (374-375)".

2. Lines 155-156: "unlike adults, ToM and pain networks were not significantly anti-correlated in three year olds ($t(16)=2.6$, $p=1$).". It's unclear to me what test is being performed. It sounds like a test of the (anti-)correlation value against zero.

My concern here is that it is not at all easy to interpret the magnitudes of correlations relative to zero in functional connectivity analyses. As discussed by many methodologists in the literature, e.g., Weissenbacher et al. 2009 NeuroImage, global signal regression (and related procedures such as CompCor) induces/enhances anti-correlations between regions; for the same reasons, the magnitude of positive correlations may shift after this procedure. In fact, interpretation of absolute magnitudes (of between-region correlations) is questionable even if no global signal regression has been performed. Thus it is best to emphasize *relative magnitudes* of correlations (e.g., ToM within-network correlation is higher than Pain within-network correlation, or ToM network vs. pain network is more negatively correlated for adults than children) and avoid drawing conclusions based on whether correlation magnitudes are large relative to zero, whether negative or positive. Fortunately, the main results of the paper do not appear to rest on tests against zero. However, there are some places where the language suggests conclusions drawn from such tests. For example, lines 123-127; 145-147; 153-156; 317-319.

It's appropriate to report correlation magnitudes (how else would you describe the data?), and I don't feel it's a problem to even report these p-values if you wish (though some researchers would say p-values are misleading under these circumstances), or to use the term "anti-correlation" as there is a conceptual usefulness to it; as long as care is taken not to over-interpret the raw magnitudes. The reader should be given a sense of which tests to take most seriously, and which are more descriptive.

3. It would be helpful to see the full scatter plots for the analyses that are currently summarized by year in Figure 1. In other words, the scatter plots associated with lines 131 & 132. Perhaps this could be a supplementary figure.

4. Line 167: "We additionally tested for neural differences based on performance on explicit false belief tasks, among 3- to 5-year-old children. "

As I understand it, the explicit false belief tasks are a subset of the questions in the battery,

the "strongest possible version" of the battery. It would be good to mention this here, as otherwise readers might mistakenly think this is an independent behavioral test from the battery.

5. Line 226: "The average timecourse in ToM and pain regions in children was highly correlated with that of adults ($r > .8$, $p < .000001$)."

It took me a while to figure out exactly what analysis was performed here. At first I thought it was timecourses averaged within year, but realized that it couldn't be that because later age=3 is reported as $r = 0.3$. I concluded that it must be two average timecourses, one for each ToM and Pain. Please report both of the ToM and Pain correlations in this analyses (children vs. adult), and please also report the same for each year.

6. Lines 232-233: "The two ToM events that showed greater responses with age are longer events that involve multiple and more complicated mental states."

This statement seems like it needs more quantitative support.

7. In Figure 2, P05 does not appear to overlap with the nearest 3-yo event. Does it overlap by just one timepoint? It would be helpful to add a few words about what is meant by the word "identified" (line 243), otherwise the reader may be confused when trying to compare this text to what is displayed in Figure 2.

8. "These events correspond to a subset of the timepoints during which the response was above baseline"

I found "baseline" hard to understand in this context until I read over the Methods in detail. Perhaps "... timepoints that were identified as Pain/ToM events" would be easier on the reader.

9. Line 262: "We found that the maturity of the movie-driven timecourse in both ToM and Pain networks was predicted by the anti-correlation of regions across networks."

Please be sure to address this analysis when considering my comment #1, as I believe it is subject to the same concern.

Reviewer #2 (Remarks to the Author):

The present study examines the developmental of neural networks for processing two different types of internal states – mental states and pain – and investigates how changes in these networks relate to changes in performance on "theory of mind" (ToM) tasks. Results indicated that ToM networks and pain networks were already distinct by age 3, but that within-network specialization and between-network differentiation continued throughout childhood. In the youngest age group (3-5 years), there were no neurological differences (either within the ToM network or between networks) between children who passed or failed the classic false-belief task.

This study deals with an important and timely topic, and it will no doubt be of considerable interest to the field. The manuscript is clear and well-written. Characterizing the

development of these networks across such a broad age-span using the same task constitutes a valuable contribution to the field. The use of this particular animated film to assess the neural correlates of ToM and pain processing is also quite clever. Because the film is entirely nonverbal and participants are not required to produce a response, it provides a way of assessing ToM processing without the verbal/response demands typically present in ToM tasks.

However, one aspect of the predictions/conclusions could use clarification. In both the introduction and the general discussion, the authors outline three different broad accounts of ToM development and state that these different accounts predict different profiles of development of ToM brain regions. The authors do not spell out what those predictions are, and this makes it somewhat difficult to evaluate whether the results support the authors' conclusions with regards to this issue.

First, the authors claim that the lack of neurological differences between the 'passers' and 'failers' of the false-belief task is inconsistent with theories that propose a 'privileged discontinuity' in ToM development. But there are several such theories and it is not clear that they all generate the same predictions for this task or that they would all predict a difference between passers/failers. For instance, Apperly and colleagues have argued that humans possess two systems for processing others behavior, one of which emerges around age 4 resulting in a discontinuity in ToM reasoning/false-belief performance. According to this account, the false-belief task and the spontaneous task performed in the scanner assess different systems and thus it is perhaps unsurprising that differences on one do not translate to differences on the other. It might be that the authors' results instead speak against a different discontinuity account that would predict neurological differences between passers/failers. If so, this should be made clear.

Second, the authors describe two different types of 'continuity' accounts. According to one, ToM develops 'slowly and continuously throughout childhood.' According to the other 'changes in executive function (e.g., response inhibition) reveal children's previously existing ToM, which does not itself undergo any change.' (As an aside, I do not think the latter sentence is a fair description of this account – arguing that the capacity to represent beliefs is present early in life is not the same as stating that ToM does not undergo any change.) The authors claim that their results support continuity (p. 19), but it is not clear whether their results distinguish between these two continuity accounts.

In addition to the discussion of continuity/discontinuity, several smaller points require clarification:

1. Children's nonverbal IQ was measured, but it does not appear that this variable was ever analyzed. Does nonverbal IQ relate to ToM performance? Did any of these analyses control for IQ? Could the age-related changes be due in part to changes in IQ rather than ToM processing per se?

2. What counted as a 'correct' response to the explanation items? Is this information provided in a prior study using these measures? If not, could this be provided in the

supplementary material?

3. The graphs show age in bins (3 years, 4 years, etc). In the analyses, was age a continuous or categorical variable?

Reviewer #3 (Remarks to the Author):

Richardson et al.. report a developmental neuroscience study of 122 children (age 3-12 yrs), who watch a movie. The movie, a short comic story, includes events that are about bodily sensations (pain etc) and/or mental states (emotions, attributions), which makes it a good stimulus to examine the neural response (and their age-typicality) in a very engaging way. The main findings are that i) classical networks associated with pain and mentalizing are detected by age 3 and ii) functional specialization increases through childhood.

This is a very important and -to my knowledge- so far the most comprehensive neuro-developmental study that uses ecologically more valid movies as stimuli. Compared to related previous work.g. on autistic individuals (Hasson, Gleran) and normal development (Cantlon), this study excels by a large sample size and a new approach to use age-appropriate narrative media as experimental tool.

The functional connectivity analyses are carried out in a state-of-the-art fashion. The results are reported adequately, including an adequate discussion of potential issues regarding to motion differences between age group, the limits of normalization for younger kids. Lastly, they have the courage to state that results should be considered exploratory, which is totally ok to do and laudable. The idea to make the data available is an excellent one (all data - or just the data to reproduce?).

While I generally found great merit in the manuscript, I had some issues to understand the analytical approach. This does not mean it is wrong - I just had difficulties following. For example, why didn't you use inter-subject correlation analysis right away? The maturity-calculation - relating the data to adults - is very close to this, and you certainly are aware of the method. However, you focus on the connectivity of networks, then use more regional reverse-correlations (which are fancy, but a bit sandwiched between the rest), and finally the maturity analyses. This all can be done, but it appears a little piecemeal and less integrated than it perhaps could be. I am sure you are aware of the study by Simony et al (from hasson group), which integrates ISC with connectivity and has some implications for your analyses as well. I am not saying you should re-run these analyses, but it would help me to better motivate/describe what the different methods do and why they were chosen. I was a bit overwhelmed and I consider myself an expert when it comes to natural stimuli and fMRI. Overall, my main concern with the article is thus not the findings or the analyses, which are fine, but rather that the structure/flow of the information could be improved to make the manuscript more readable to a broader audience.

Overall, great work!

Response to Referees

Overall we would like to thank the reviewers for their attentive reading of our paper, and thoughtful evaluations. Responding to these reviewers has helped to improve the manuscript, and has been a rewarding experience for all of us.

Reviewer #1 (Remarks to the Author):

The questions and analysis approach of the study are very interesting, and the dataset is impressive. The manuscript is well-written and the analyses are clearly described. I believe the results will be of interest to many researchers, and the public release of the dataset would benefit the community.

We thank the reviewer for this positive evaluation of our contribution.

1. In adults, ToM and Pain networks are 1) strongly correlated within-network and 2) negatively correlated across network, as previously observed in the literature. A main result of the current paper is that both effects strengthen as age increases.

Network ROIs are defined from an independent adult group and then placed onto the MNI-aligned child brain. Is it possible that the above result (effects strengthening with age) is due to increasingly better match (coregistration) between the child brain and adult brain with increasing child age?

The reviewer here seems to be describing two possible concerns. Here the reviewer seems to be concerned about the quality of the registration: could the fit somehow be noisier / less accurate in children? Registration of each individual's brain to the MNI template was manually checked during the pre-processing, including checking the match of the cortical envelope and internal features like the AC and PC and major sulci. Thus we are confident that the registrations are accurate, across all age groups. We have added the fact that we visually inspected each child's registration to the methods section (lines 478-480):

“Registration of each individual's brain to the MNI template was visually inspected, including checking the match of the cortical envelope and internal features like the AC-PC and major sulci.”

However, that leaves the second concern, as described below: even given accurate registration, children's small brains are warped more (and upsampled differentially) to fit an adult template. See below for our response to this second concern.

I understand that there is some nuance to this question; children's brains are surely becoming anatomically more like adult brains with age, and this problem is deeply entangled with the goal of understanding how children's brains are becoming functionally more like adult brains.

However, I think this is a legitimate possible confound that should be addressed.

Some ideas for approaching the problem:

- Run a simulation with a separate adult dataset as I described above, adding varying amounts of noise to coregistration parameters (or some equivalent manipulation of noise).

- Present a control network that does not show the pattern of increasing within-network correlation with increasing age. Note that in order to be a good comparison, this control network should be comparable to ToM and Pain in its adult within-network correlation.
- Put a functional connectivity seed in PCC for each child brain and calculate how much the map overlaps (Dice or Jaccard index) with the same PCC-seed maps in the adult brain and with other subjects of the same age. This can be plotted out for different correlation thresholds, and could help give a sense for whether network variability is substantially higher in the child brain in general.

We thank the reviewer for the constructive review! We decided that the second option most directly addresses the reviewer's concern.

In brief: option 1, simulating noisy registration in adult data, might not fully address the reviewer's concern, because we might not simulate the right kind of noisy registration; option 3, using PCC seeds as a 'reality check', seems like it also might not give a definitive result, since the placement of the seed is subject to the same concerns about imperfect registration as the placement of our ROIs. Option 2 therefore offers the most direct test of this concern, by using the same real data as our main analyses.

We have added a figure to the supplemental materials (Supplementary Figure 4), which corresponds to the second suggestion above. We calculated within-functional-network correlations for four pairs of regions: bilateral face regions, scene regions, primary motor cortex and primary visual cortex, as well as the across-network correlations (across-face-scene; across-M1-V1). As is visible in the figure, within-V1 and within-face region correlations are similar in magnitude to the within ToM and pain network correlations in children, on average, but do not change with age (see figure legend). This shows that mature within-network correlations can be measured even in our youngest children. Further, the across-face-scene and across-M1-V1 correlations do not change with age, and these networks are not anti-correlated at any age. These results suggest that anti-correlations across-networks are not artifacts of data quality, registration quality or data analysis steps, in any age. On the other hand, we did find marginal changes with age between bilateral scene regions, and a significant change with age between bilateral M1. This effect was unpredicted (especially since our experiment involved passive viewing), and might be an interesting target for future research. On the other hand, it might also reflect some general effect of noise or data quality on a subset of inter-regional correlations. So for now, to ensure that we are reporting conservative estimates of change with age in our predicted, target networks, we included the average within-M1 correlation as a nuisance regressor in analyses of the effect of age on the target networks.

2. Lines 155-156: "unlike adults, ToM and pain networks were not significantly anti-correlated in three year olds ($t(16)=2.6, p=1$)." It's unclear to me what test is being performed. It sounds like a test of the (anti-)correlation value against zero.

My concern here is that it is not at all easy to interpret the magnitudes of correlations relative to zero in functional connectivity analyses. As discussed by many methodologists in the literature, e.g., Weissenbacher et al. 2009 NeuroImage, global signal regression (and related procedures

such as CompCor) induces/enhances anti-correlations between regions; for the same reasons, the magnitude of positive correlations may shift after this procedure. In fact, interpretation of absolute magnitudes (of between-region correlations) is questionable even if no global signal regression has been performed. Thus it is best to emphasize *relative magnitudes* of correlations (e.g., ToM within-network correlation is higher than Pain within-network correlation, or ToM network vs. pain network is more negatively correlated for adults than children) and avoid drawing conclusions based on whether correlation magnitudes are large relative to zero, whether negative or positive.

Fortunately, the main results of the paper do not appear to rest on tests against zero. However, there are some places where the language suggests conclusions drawn from such tests. For example, lines 123-127; 145-147; 153-156; 317-319.

It's appropriate to report correlation magnitudes (how else would you describe the data?), and I don't feel it's a problem to even report these p-values if you wish (though some researchers would say p-values are misleading under these circumstances), or to use the term "anti-correlation" as there is a conceptual usefulness to it; as long as care is taken not to over-interpret the raw magnitudes. The reader should be given a sense of which tests to take most seriously, and which are more descriptive.

Thank you for this very reasonable feedback. We have removed the text that reported significance above or below a baseline of zero. We have kept the qualitative descriptions of these correlations (e.g. “By contrast, unlike adults, ToM and pain networks were not anti-correlated in three year olds (Across-network correlation=.05(.02))”), and have emphasized the comparison between within and across network correlations (e.g. lines 144-145). We have kept the numerical values for within and across correlations separate, because we feel it is informative: reporting change in the difference between within - across correlations would make it less clear how much each measure (within-network correlation and across-network correlation) is changing.

Relatedly, we believe that the inclusion of the control networks added to the supplemental materials (described above) helps to strengthen the claim that ToM and pain networks become more anti-correlated with age, given that these control networks are not anti-correlated (at any age).

3. It would be helpful to see the full scatter plots for the analyses that are currently summarized by year in Figure 1. In other words, the scatter plots associated with lines 131 & 132. Perhaps this could be a supplementary figure.

We have added this figure to the supplemental materials as requested- see Supplementary Figure 3.

Supplementary Figure 3. Inter-regional correlations by age. Within-ToM (red, left), Within-Pain (green, middle), and across-ToM-Pain network correlations, by age (x-axis), in all children (n=122).

4. Line 167: "We additionally tested for neural differences based on performance on explicit false belief tasks, among 3- to 5-year-old children. "

As I understand it, the explicit false belief tasks are a subset of the questions in the battery, the "strongest possible version" of the battery. It would be good to mention this here, as otherwise readers might mistakenly think this is an independent behavioral test from the battery.

You are correct in that the explicit false-belief questions are a subset of the questions in the ToM battery. We have clarified this in lines 164-165:

"We additionally tested for neural differences based on performance on explicit false belief questions, among 3- to 5-year-old children. These questions were a subset of the questions in the ToM behavioral battery (see Methods)."

5. Line 226: "The average timecourse in ToM and pain regions in children was highly correlated with that of adults ($r_s > .8$, $p_s < .000001$)."

It took me a while to figure out exactly what analysis was performed here. At first I thought it was timecourses averaged within year, but realized that it couldn't be that because later age=3 is reported as $r=0.3$. I concluded that it must be two average timecourses, one for each ToM and Pain. Please report both of the ToM and Pain correlations in this analyses (children vs. adult), and please also report the same for each year.

We have edited the statistics reported to reflect the correlation between the average adult ToM and pain timecourse and the average timecourse in these networks for each age group among the children (lines 215-217).

"The average timecourse in ToM and pain regions in children was highly correlated with that of adults (ToM: 3yo: $r=.30$, 4yo: $r=.33$, 5yo: $r=.65$, 7yo: $r=.76$, 8-12yo: $r=.85$ (all $p_s < .0002$); Pain: 3yo: $r=.60$, 4yo: $r=.57$, 5yo: $r=.74$, 7yo: $r=.84$, 8-12yo: $r=.90$ (all $p_s < .000001$))."

6. Lines 232-233: "The two ToM events that showed greater responses with age are longer events that involve multiple and more complicated mental states."

This statement seems like it needs more quantitative support.

While the length of the events is easy to quantify, the limits of the stimulus make it difficult to quantify the way in which these two events involve more complicated mental state reasoning. We have provided more comprehensive descriptions of all of the events in Supplementary Table 3; this primarily involved adding more content to these two events (given that all of the original descriptions were short; and these are two are longer events). We have additionally provided a link to Pixar's website (Line 430-432), which has a short description of the plot of the movie:

"Participants watched a silent version of "Partly Cloudy," (Pixar Animation Studios), a 5.6-minute animated movie. A short description of the plot can be found online (<https://www.pixar.com/partly-cloudy#partly-cloudy-1>)."

7. In Figure 2, P05 does not appear to overlap with the nearest 3-yo event. Does it overlap by just one timepoint? It would be helpful to add a few words about what is meant by the word "identified" (line 243), otherwise the reader may be confused when trying to compare this text to what is displayed in Figure 2.

We are extremely grateful to the reviewer for highlighting this, as the mismatch between the text and figure reflected an error in the text. We have corrected the text and made the description more clear (see below for lines 233-240). Thank you again for reading so carefully.

"Reverse correlation analysis conducted on the three year olds' data alone identified four of the twelve pain events and one of the seven ToM events discovered in the adult sample. These events correspond to a subset of the timepoints that were identified as ToM or pain events in three year olds (Pain: 14/32s, ToM: 4/8s). Interestingly, 8 of the remaining 18s identified as a pain event in three-year-old children corresponds to a ToM event (T04) in adults, and the remaining 4s identified as a ToM event corresponds to a pain event (P01) in adults (see Figure 2). The remaining 10s identified as pain events occurred immediately after adult pain event timepoints."

8. "These events correspond to a subset of the timepoints during which the response was above baseline"

I found "baseline" hard to understand in this context until I read over the Methods in detail.

Perhaps "... timepoints that were identified as Pain/ToM events" would be easier on the reader.

We have edited this sentence as suggested (Lines 235-236):

"These events correspond to a subset of the timepoints that were identified as ToM or pain events in three year olds (Pain: 14/32s, ToM: 4/8s)."

9. Line 262: "We found that the maturity of the movie-driven timecourse in both ToM and Pain networks was predicted by the anti-correlation of regions across networks."

Please be sure to address this analysis when considering my comment #1, as I believe it is subject to the same concern.

We believe that the added supplemental figure (Supplementary Figure 4) addresses this concern, as it shows that while ToM and pain networks become increasingly anti-correlated with age, other comparable across-network correlations do not show change with age, and are not anti-correlated among adults (across-face-scene correlations, across-M1-V1 correlations).

Reviewer #2 (Remarks to the Author):

This study deals with an important and timely topic, and it will no doubt be of considerable interest to the field. The manuscript is clear and well-written. Characterizing the development of these networks across such a broad age-span using the same task constitutes a valuable contribution to the field. The use of this particular animated film to assess the neural correlates of ToM and pain processing is also quite clever. Because the film is entirely nonverbal and participants are not required to produce a response, it provides a way of assessing ToM processing without the verbal/response demands typically present in ToM tasks.

We thank the reviewer for this positive evaluation!

However, one aspect of the predictions/conclusions could use clarification. In both the introduction and the general discussion, the authors outline three different broad accounts of ToM development and state that these different accounts predict different profiles of development of ToM brain regions. The authors do not spell out what those predictions are, and this makes it somewhat difficult to evaluate whether the results support the authors' conclusions with regards to this issue.

First, the authors claim that the lack of neurological differences between the 'passers' and 'failers' of the false-belief task is inconsistent with theories that propose a 'privileged discontinuity' in ToM development. But there are several such theories and it is not clear that they all generate the same predictions for this task or that they would all predict a difference between passers/failers. For instance, Apperly and colleagues have argued that humans possess two systems for processing others behavior, one of which emerges around age 4 resulting in a discontinuity in ToM reasoning/false-belief performance. According to this account, the false-belief task and the spontaneous task performed in the scanner assess different systems and thus it is perhaps unsurprising that differences on one do not translate to differences on the other. It might be that the authors' results instead speak against a different discontinuity account that would predict neurological differences between passers/failers. If so, this should be made clear.

We thank the reviewer for raising this important point. Interestingly, the existing neuroscience evidence appears inconsistent with the idea that a different set of brain

regions is recruited to reason about nonverbal, implicit or spontaneous ToM tasks, as compared to verbal, explicit or prompted ToM tasks. For example, the current study uses a spontaneous-viewing stimulus, which has been validated as a localizer for regions recruited to process explicit, verbal false-belief narratives (Jacoby et al., 2016). Other work also finds that nonverbal tasks (Gallagher et al., 2000; Kobayashi et al., 2007) and more traditional spontaneous or implicit tasks (e.g. those initially used in behavioral paradigms designed to measure implicit ToM) also seem to recruit this same set of ToM brain regions (e.g. Sommer et al., 2007; Schneider et al., 2014).

This set of findings supports the hypothesis that if there *are* neural differences between children based on theory of mind behavior, we would expect these differences to arise within this same set of ToM brain regions. We have edited the introduction to make our predictions more clear (lines 31-45, see below). We have also edited the discussion to highlight ways in which our results are consistent or inconsistent with each of these developmental hypotheses (lines 326-398).

Second, the authors describe two different types of ‘continuity’ accounts. According to one, ToM develops ‘slowly and continuously throughout childhood.’ According to the other ‘changes in executive function (e.g., response inhibition) reveal children’s previously existing ToM, which does not itself undergo any change.’ (As an aside, I do not think the latter sentence is a fair description of this account – arguing that the capacity to represent beliefs is present early in life is not the same as stating that ToM does not undergo any change.) The authors claim that their results support continuity (p. 19), but it is not clear whether their results distinguish between these two continuity accounts.

We hope that the edits to the introduction have made our argument more clear (lines 31-45):

“Based on theories in developmental psychology, we derive three predictions for observations in the social brain regions of young children. First, success on explicit false belief tasks could reflect an important conceptual leap or discontinuity in ToM development, as theories of others’ internal states are dramatically altered by insight into the representational nature of mental states (28, 29). On this view, the division between cortical responses to others’ bodies versus minds might emerge just when children explicitly understand false beliefs. Second, success on explicit false belief tasks could reflect development in other domain-general brain regions, removing earlier performance limitations (e.g. (30-32)). On this view, spontaneous processing of others’ mental states within domain-specific regions for ToM might be similar, regardless of performance on explicit tasks. Third, success on explicit false belief tasks could be just one step in ongoing conceptual development within ToM, which begins before and continues after false belief reasoning (33-36). On this view, change within ToM brain regions might occur both before and after passing explicit false belief tasks. Of course, these predictions are only some of the possibilities that could be derived from each theoretical perspective; and reality could include a mixture of these three views.”

In addition to the discussion of continuity/discontinuity, several smaller points require

clarification:

1. Children's nonverbal IQ was measured, but it does not appear that this variable was ever analyzed. Does nonverbal IQ relate to ToM performance? Did any of these analyses control for IQ? Could the age-related changes be due in part to changes in IQ rather than ToM processing per se?

Children's nonverbal IQ was measured using two different measures: children younger than age five completed the WPPSI block design task, whereas children five years and older completed the KBIT-2 matrix reasoning task (see lines 625-627). This was a consequence of younger and older children initially being recruited for different studies (see lines 616-619); additionally, the WPPSI matrix reasoning task is not standardized for children younger than age four (hence the choice to use the block design task). Because these tasks are not comparable, we did not include these analyses in the manuscript.

For the reviewer's interest, we conducted the suggested analyses separately for each group (grouped according to the nonverbal IQ measure that was collected).

Within the younger subset of participants (ages 3-4 years, n=31), age and raw IQ are similar in their effect on ToM performance (effect of age: $b=.31$, $t=1.8$, $p=.08$, effect of IQ: $b=.32$, $t=1.9$, $p=.07$), whereas in the older subset of participants (ages 5-12 years, n=91), age is a stronger predictor of ToM performance than raw IQ (effect of age: $b=.5$, $t=4.5$, $p<.00002$, effect of IQ: $b=.2$, $t=1.8$, $p=.07$).

Standardized IQ does not change with age (young: $r=-.06$, $p=.7$; old: $r=.19$, $p=.08$), and is a worse predictor of ToM performance than age in both subsets of participants (young: effect of age: $b=.43$, $t=2.7$, $p=.01$, effect of IQ: $b=.29$, $t=1.8$, $p=.09$; old: effect of age: $b=.7$, $t=8.8$, $p<.0000001$, effect of IQ: $b=.1$, $t=1.3$, $p=.2$).

In sum, the developmental change in ToM performance observed in our sample is better predicted by age than by nonverbal IQ.

We did collect a measure of response inhibition among all three to five year old children- the Dimensional Change Card Sort (DCCS) task. Children older than five years of age are expected to perform at ceiling to this task (Zelazo et al., 2006). Response inhibition is hypothesized to be particularly important for passing explicit false-belief tasks (Carlson et al., 2004); in our data DCCS performance and ToM score are positively correlated, controlling for age (see lines 89-91):

"In the three to five-year-old subset of children who completed both measures (n=64), ToM and DCCS scores were positively correlated (partial correlation controlling for age: $r_k(61)=.19$, $p=.03$)."

To address the reviewer's concern, we therefore tested whether the observed association between false belief performance, and within-ToM correlation strength, in 3-5 year old children was also associated with DCCS performance. We found that once we include

DCCS in the regression, the effect of false-belief on within-ToM correlation falls below the criterion for significance. Age remains the strongest predictor of within-ToM correlation strength; DCCS itself does not have a significant effect on within-ToM correlation strength, so appears to be reducing the predictive power of false belief performance because of sharing variance between correlated regressors (see lines 165-172):

“There was a significant difference in within-ToM network correlation between explicit false belief task “passers” and “failers” (Within-ToM: M(SE) Passers: .29(.02), Failers: .25(.03), effect of group (pass vs. fail): $b=-.70$, $t=-2.06$, $p=.046$, effect of age: $b=.73$, $t=4.4$, $p<.0005$, effect of motion: $b=-.34$, $t=-2.7$, $p=.009$). This group difference becomes marginal when response inhibition (DCCS summary score) is additionally included in the regression (effect of group (pass vs. fail): $b=-.64$, $t=-1.80$, $p=.079$, effect of age: $b=.74$, $t=4.4$, $p<.0001$, effect of motion: $b=-.33$, $t=-2.5$, $p=.02$, NS effect of DCCS (response inhibition): $b=-.08$, $t=-.59$, $p=.56$).”

Because the original group effect was already quite weak ($p=.046$), and because we did not previously interpret it strongly, we do not believe that these new results change the overall structure or import of the manuscript. However, we do feel it important to report the full set of analyses and results. We thank the reviewer for the suggestion!

Additionally, we now include a table in the supplementary materials (Supplementary Table 1) that provides information about each age group (n, gender, handedness, M(SD) and range of age, M(SD) of nonverbal IQ, M(SD) of ToM score, and (when applicable) M(SD) of DCCS Summary.

2. What counted as a ‘correct’ response to the explanation items? Is this information provided in a prior study using these measures? If not, could this be provided in the supplementary material?

We have made the booklet task (story script, example videos of administration, and coding instructions and examples) available on the Open Science Framework website (<https://osf.io/G5ZPV/>; DOI: 10.17605/OSF.IO/G5ZPV; ARK: c7605/osf.io/g5zpv). The coding examples include examples of correct and incorrect explanations for false belief, moral reasoning, interpretation, and reference questions. The availability of these materials is now referenced in the “Data availability” section.

3. The graphs show age in bins (3 years, 4 years, etc). In the analyses, was age a continuous or categorical variable?

Age was used as a continuous variable in all analyses. We have clarified this in the methods section as well as in figure legends, and have additionally added a figure to the supplemental materials that shows the interregional correlation analysis measures (within-network and across-network correlations) by age (Supplementary Figure 3).

Lines 545-551

“In order to test for developmental change in within- and across-network correlations, we conduct linear regressions to test for 1) significant differences between adults and children, in regressions that include group (child vs. adult) and number of artifact timepoints as predictors, and 2) significant effects of age (as a continuous variable), ToM performance, and number of artifact timepoints among children, and 3) significant group differences between children who pass and fail explicit false belief tasks, including number of artifact timepoints and age as predictors.”

Lines 587-591

“In order to test for developmental effects in the magnitude of response to ToM and pain events, we defined a peak timepoint per event as the timepoint with the highest average signal value in adults, and tested for significant correlations between magnitude of response at peak timepoints and age (as a continuous variable) and (in the case of ToM regions only) ToM performance.”

Reviewer #3 (Remarks to the Author):

This is a very important and -to my knowledge- so far the most comprehensive neuro-developmental study that uses ecologically more valid movies as stimuli. Compared to related previous work e.g. on autistic individuals (Hasson, Gleran) and normal development (Cantlon), this study excels by a large sample size and a new approach to use age-appropriate narrative media as experimental tool.

We thank the reviewer for this positive evaluation!

The functional connectivity analyses are carried out in a state-of-the-art fashion. The results are reported adequately, including an adequate discussion of potential issues regarding to motion differences between age group, the limits of normalization for younger kids. Lastly, they have the courage to state that results should be considered exploratory, which is totally ok to do and laudable. The idea to make the data available is an excellent one (all data - or just the data to reproduce?).

All of the data. We plan to share the raw fMRI data, the preprocessed fMRI data, anatomical images, the fMRI paradigm information, and the behavioral and demographics data. We have updated our data availability plan to be more clear (lines 670-674):

The fMRI and behavioral data collected and analyzed during the current study are available through the OpenfMRI project (<https://openfmri.org/>; [data specific link forthcoming]). The ToM behavioral battery is additionally available through OSF (<https://osf.io/G5ZPV/>; DOI: 10.17605/OSF.IO/G5ZPV; ARK: c7605/osf.io/g5zpv). The corresponding author welcomes any additional requests for materials or data.

While I generally found great merit in the manuscript, I had some issues to understand the analytical approach. This does not mean it is wrong - I just had difficulties following. For example, why didn't you use inter-subject correlation analysis right away? The maturity-calculation - relating the data to adults - is very close to this, and you certainly are aware of the

method. However, you focus on the connectivity of networks, then use more regional reverse-correlations (which are fancy, but a bit sandwiched between the rest), and finally the maturity analyses. This all can be done, but it appears a little piecemeal and less integrated than it perhaps could be. I am sure you are aware of the study by Simony et al (from Hasson group), which integrates ISC with connectivity and has some implications for your analyses as well. I am not saying you should re-run these analyses, but it would help me to better motivate/describe what the different methods do and why they were chosen.

I was a bit overwhelmed and I consider myself an expert when it comes to natural stimuli and fMRI. Overall, my main concern with the article is thus not the findings or the analyses, which are fine, but rather that the structure/flow of the information could be improved to make the manuscript more readable to a broader audience.

Overall, great work!

Thank you!

We have edited the introduction to motivate and describe the methods we chose for the timecourse analyses, in order to make the manuscript easier to read (see Lines 63-80; 180-182).

We have additionally referenced the relevant article by Simony et al. (2016) in the discussion section; we agree this approach is interesting, relevant, and promising!

Lines 315-318:

“An advantage of measuring inter-regional correlations during a movie is that children's psychological state (e.g. attention, anxiety, alertness) is likely more similar, across ages. On the other hand, a disadvantage is that we cannot distinguish between intrinsic and task-driven contributions to the inter-regional correlations{Simony:2016wf}.”

Reviewers' comments:

Reviewer #1 (Remarks to the Author):

Thank you for conducting the control analyses with Face-Scene and M1-V1 networks, it is very helpful. Overall I think the authors have done a great job of addressing the reviewers' questions and the paper is much improved. There are a few things about the new analyses that I have remaining questions about.

I wouldn't mind also seeing the scatter plots and fitted regression lines for the data in Fig S4, as opposed to having it broken up by year, as these analyses were not conducted by year bins. As it is, it's rather hard to see the effect in question: that Tom-Pain-within is going up with age and Tom-Pain-across is going down with age, while the same is not true for Face-Scene and for M1-V1. Perhaps this would even be better displayed by plotting the residuals after regressing out motion. For example, the result for Across-M1-V1 is $r=.18$, $p=.05$, but this isn't readily apparent when looking at the light blue box plots across years in the bottom row of the figure. The visualization is a minor thing though; I leave it up to the authors.

Within-network correlation changes across age for ToM and Pain, and across-network correlation changes across age for ToM-Pain. The same is not true for Face-Scene and for M1-V1. Is there a significant interaction between ToM-Pain and Face-Scene? Between Tom-Pain and M1-V1? I believe some readers will want to know this; for example, if you were reporting ROI activation levels, showing that an effect is significant in one region but not significant in a control region typically includes reporting the region interaction. Personally I am not too concerned about the statistical thresholds as long as the data are displayed well and effect sizes are reported, but some readers might feel this is missing.

An additional analysis that I would very much like to see is the "maturity" analyses (correlation between children and adult timecourses) in the control networks. I.e., the correlations reported on lines 274-276, for Face, Scene, M1, and V1 networks.

The M1 timecourse is used as a regressor for inter-region correlation analyses, both in the original manuscript and the revision. However, when the analyses are described in the main text, this covariate is described as "motion" or "effect of motion" (ln 188 and throughout this section). I think it's worth mentioning at least once in the main text that this is the M1 timecourse, as I believe most readers will assume you are referring to the head motion parameters typically calculated during preprocessing.

Were M1 timecourses regressed out for the reverse correlation analyses and/or for the "maturity" analyses (correlations between children and adults, lines 274-276)? The Methods say that the regression was conducted "for inter-region correlation analyses", suggesting that the answer is no, but I just want to clarify.

Reviewer #2 (Remarks to the Author):

This revision has addressed all of my comments/concerns regarding the previous draft of the manuscript. The paper is well written and the conclusions are sound. These data and this clever, ecologically-valid method for assessing processing of others' minds and bodies constitute a strong contribution to the field. I have no further suggestions for the authors.

Reviewer #3 (Remarks to the Author):

This is a great study and the revised manuscript has further improved the quality of the paper. I have no further requests.
Great work!

Response to Referees

Reviewer #1 (Remarks to the Author):

Thank you for conducting the control analyses with Face-Scene and M1-V1 networks, it is very helpful. Overall I think the authors have done a great job of addressing the reviewers' questions and the paper is much improved.

Thank you!

There are a few things about the new analyses that I have remaining questions about.

I wouldn't mind also seeing the scatter plots and fitted regression lines for the data in Fig S4, as opposed to having it broken up by year, as these analyses were not conducted by year bins. As it is, it's rather hard to see the effect in question: that Tom-Pain-within is going up with age and Tom-Pain-across is going down with age, while the same is not true for Face-Scene and for M1-V1. Perhaps this would even be better displayed by plotting the residuals after regressing out motion. For example, the result for Across-M1-V1 is $r=.18$, $p=.05$, but this isn't readily apparent when looking at the light blue box plots across years in the bottom row of the figure. The visualization is a minor thing though; I leave it up to the authors.

We have added scatter plots that show the within- and across- network correlation r -values from the expanded interregional correlation analysis to the supplementary information (Supplementary Figure 3; included at the end of this document). These correlation values are based on the timecourses without regressing out the bilateral M1 timecourse, and are not fisher z -transformed.

Within-network correlation changes across age for ToM and Pain, and across-network correlation changes across age for ToM-Pain. The same is not true for Face-Scene and for M1-V1. Is there a significant interaction between ToM-Pain and Face-Scene? Between Tom-Pain and M1-V1? I believe some readers will want to know this; for example, if you were reporting ROI activation levels, showing that an effect is significant in one region but not significant in a control region typically includes reporting the region interaction. Personally I am not too concerned about the statistical thresholds as long as the data are displayed well and effect sizes are reported, but some readers might feel this is missing.

We have added the following to the supplemental information; see Supplementary Figure 4 Legend:

“Positive correlations between within-ToM and within-Pain correlations and age were significantly stronger than within-Face and within-V1 correlations, but not significantly stronger than within-Scene and within-M1 correlations (test differences in age correlations: Within-Face: vs. within-ToM: $z=2.69$, $p=.01$, vs. within-Pain: $z=2.66$, $p=.01$; within-Scene: vs. within-ToM: $z=1.33$, $p=.18$, vs. within-Pain: $z=1.3$, $p=.19$; M1: vs. within-ToM: $z=.76$, $p=.44$, vs. within-Pain: $z=.73$, $p=.47$; V1: vs. within-ToM: $z=4.35$, $p=0$, vs. within-Pain: $z=4.32$, $p=0$). The across ToM-Pain anti-correlation was

significantly stronger than the across Face-Scene and across M1-V1 anti-correlations (Face-Scene: $z(122)=2.2$, $p=.03$; M1-V1: $z(122)=3.39$, $p=0$)."

An additional analysis that I would very much like to see is the "maturity" analyses (correlation between children and adult timecourses) in the control networks. I.e., the correlations reported on lines 274-276, for Face, Scene, M1, and V1 networks.

We have added this information to the supplemental information; see Supplemental Table 3 (included at the end of this document).

The M1 timecourse is used as a regressor for inter-region correlation analyses, both in the original manuscript and the revision. However, when the analyses are described in the main text, this covariate is described as "motion" or "effect of motion" (ln 188 and throughout this section). I think it's worth mentioning at least once in the main text that this is the M1 timecourse, as I believe most readers will assume you are referring to the head motion parameters typically calculated during preprocessing.

The M1 timecourse was regressed out for the inter-region correlation analysis reported in the main text (and not the expanded inter-region correlation analysis in the supplementary information). For all correlation and regression analyses (analyses conducted to test for change with age, or differences between groups), the amount of motion, as measured by the number of artifact timepoints, was included as a covariate. We use the word "motion" to mean "head motion in the scanner" only, and not to refer to the M1 timecourse. We have edited the text to clarify:

First instance of using the term "motion" in the results section: *"Among children, within-ToM and within-Pain network correlations increased significantly with age (within-ToM: $r_s(119)=.37$, $p<.00005$; within-Pain: $r_s(119)=.28$, $p=.002$; including motion (number of artifact timepoints) as a covariate)."*

Figure 2 Legend: *"Asterisk denotes a significant effect of false belief task group (pass vs. fail) in a regression that also includes age and amount of motion (number of artifact timepoints) as covariates ($p<.05$)"*

Methods (lines 500-505): *"Despite amount of motion being matched across children, and therefore likely not driving developmental effects within the child sample, we include number of motion artifact timepoints as a covariate in all analyses. Number of artifact timepoints is highly correlated with measures of mean translation, rotation, and distance ($r_s>.8$). Because this measure is not normally distributed, spearman correlations were used when including amount of motion as a covariate in partial correlations."*

Methods (lines 596-601): *"In order to test for developmental effects in the magnitude of response to ToM and pain events, we defined a peak timepoint per event as the timepoint with the highest average signal value in adults, and tested for significant correlations between magnitude of response at peak timepoints and age (as a continuous variable),"*

including amount of motion (number of artifact timepoints) as a covariate. Because this measure of motion is non-normally distributed, we employed spearman correlations.”

Were M1 timecourses regressed out for the reverse correlation analyses and/or for the "maturity" analyses (correlations between children and adults, lines 274-276)? The Methods say that the regression was conducted "for inter-region correlation analyses", suggesting that the answer is no, but I just want to clarify.

You are correct- the bilateral M1 timecourse was regressed out only for the inter-region correlation analyses. We have made this more clear in the methods section:

“For inter-region correlation analyses only, we additionally regressed out the raw timecourse extracted from bilateral primary motor cortex (MI). Primary motor cortex ROIs were 10mm spheres drawn around peak coordinates generated with Neurosynth (<http://neurosynth.org/>; search term: “primary motor,” forward inference from 273 studies; coordinates: [38,-24,58], [-38,-20,58]). These ROIs are included in the expanded inter-region correlation analysis shown in Supplementary Figure 4; the bilateral M1 timecourse was not regressed out for this supplemental analysis. However, because this analysis showed that the within-M1 inter-region correlation increases with age among children, we regressed the bilateral M1 timecourse from the ToM and Pain timecourses for the inter-region correlation analyses reported in the main text, to ensure that the age effects in the ToM and pain networks are above and beyond developmental effects present in regions like primary motor cortex, and that within-network correlations are not falsely inflated by commonalities in signal fluctuation across the brain.”

As well as in the figure legend for Supplementary Figure 4:

“The M1 timecourse is not regressed out from the timecourses analyzed for this figure/the expanded IRC analysis.”

Reviewer #2 (Remarks to the Author):

This revision has addressed all of my comments/concerns regarding the previous draft of the manuscript. The paper is well written and the conclusions are sound. These data and this clever, ecologically-valid method for assessing processing of others' minds and bodies constitute a strong contribution to the field. I have no further suggestions for the authors.

Thank you!

Reviewer #3 (Remarks to the Author):

This is a great study and the revised manuscript has further improved the quality of the paper. I have no further requests.
Great work!

Thank you!

We have made a few edits of our own, which we highlight here for ease of review:

1. We have added the following sentences, to clarify the logic behind conducting the inter-region correlation analysis on non-bilateral regions:

“The strongest within-network correlations in the three year olds were between homologous pairs of regions in opposite hemispheres, such as right and left TPJ (ToM), and the right and left insula (Pain). These strong correlations, between pairs of regions that are functionally homologous but physically distant, suggest that even the data from three year old children are of high enough quality to detect inter-region correlations when they exist; and therefore that changes with age in other inter-regional correlations reflect real changes in the functional relationships between those regions. However, the functional separation of the two networks was not fully explained by the strong correlations between bilateral pairs (M(SE) Within-non-bilateral-ToM correlation=.20(.02), Within-non-bilateral-Pain correlation=.17(.02); within-non-bilateral-ToM vs. across: $t(16)=5.1$, $p=.0001$, within-non-bilateral-Pain vs. across: $t(16)=4.4$, $p=.0005$; paired two-tailed t-test).”

2. For the functional maturity analyses, we now use raw time courses (instead of z-scored time courses), for consistency with previous papers measuring functional maturity of fMRI time courses (e.g. Canton & Li 2013). All of our claims are unaffected by this change.
3. For the analysis of the bilateral fusiform face area, in the supplementary materials, we now included all 122 participants (instead of excluding 17 participants who had partial coverage of that region). The claims of this supplementary analysis are not affected by this change.
4. We have reworded the legends of Supplementary Fig 5 & 6 to increase clarity. The new legends read:

“Supplementary Figure 5. Bilateral Fusiform Reverse Correlation Analysis. a) Bilateral fusiform regions of interest (ROIs). Regions are face parcels created and made publically available⁷⁹. b) Average timecourse of response extracted from bilateral fusiform ROIs, per age group. Shaded light orange blocks denote events identified in a reverse correlation analysis of the timecourse of response in adult participants ($n=33$; see Methods); dark orange outlines denote events identified in a reverse correlation analysis of the timecourse of response in three-year-old children ($n=17$). Event labels (e.g. F01, F02) reflect rank order of magnitude of response in adults. c) Example frame, short description, timing and duration, timepoint of peak response, and response magnitude by age group for each event identified in the reverse correlation analysis.⁴⁷ Error bars represent standard error. Peak timepoints were chosen based on the adult data, included here for illustration. Statistical tests of age-related change were computed only on data from children. The magnitude of response in bilateral fusiform does not

change with age among children (Bonferroni correction for multiple comparisons $\alpha = .0071$, correcting for 7 events/tests; $|r|s < .24$, $ps > .01$).”

“Supplementary Figure 6. Reverse Correlation Analysis: ToM and Pain events. The reverse correlation analysis in adults identified seven ToM events (top) and twelve pain events (bottom). For each event, an example frame and description are given ⁴⁷. The bar graphs show the average response magnitude of response per age group in the ToM and Pain networks, for each event. Peak timepoints were chosen based on the adult data, included here for illustration. Statistical tests of age-related change were computed only on data from children. Asterisks denote events that evoke significantly greater responses with age (black; controlling for motion and correcting for multiple comparisons (MC) (19 events, $\alpha = .0026$)), or ToM behavioral performance (red; controlling for age, motion, and correcting for MC (7 events, $\alpha = .007$)). Event labels (e.g. T01, T02) in bold type are those that were replicated in an independent sample of adults ($n=20$; Supplementary Fig. 7). See Supplementary Table 4 for event timing, duration, and full descriptions.”

We thank you again for your help in improving this manuscript!

Supplementary Figure 3

Supplementary Figure 3. Inter-regional correlations by age. Correlations are the raw (non z-scored) r -values (y-axis), calculated on the “raw” timecourses (without regression of the bilateral-M1 timecourse). Correlation values are shown for all children ($n=122$), with age on the x-axis. **Top row:** Within-ToM (red), Within-Pain (green), and across-ToM-Pain (dark blue) network correlations. **Middle row:** Within-Face (orange), Within-Scene (light blue), and across-Face-Scene (navy) network correlations. **Bottom row:** Within-M1 (purple), Within-V1 (bright blue), and across-M1-V1 (light blue) network correlations. See results (main text) and Supplementary Figure 4 for statistics on change with age (all statistical tests used z-scored correlation values).

Supplemental Table 3

Age Group	ToM	Pain	Face	Scene	M1	V1
3yo	0.28	0.60	0.75	0.61	0.11	0.53
4yo	0.31	0.56	0.59	0.67	0.06	0.61
5yo	0.60	0.73	0.71	0.77	0.35	0.78
7yo	0.72	0.83	0.84	0.80	0.44	0.74
8-12yo	0.82	0.89	0.86	0.85	0.53	0.76

Supplemental Table 3. Average timecourse correlations. This table provides the correlation value (r) between the average timecourse of response in each network included in the expanded IRC analysis, for each age group, and the corresponding average timecourse of response in adults. These timecourses are the same as those used for the reverse correlation analysis (the M1 timecourse is not included as a regressor), prior to z-normalization. All correlations are significantly positive ($p < .0005$) except those shaded in grey (3yo M1: $p = .16$; 4yo M1: $p = .46$).

REVIEWERS' COMMENTS:

Reviewer #1 (Remarks to the Author):

The authors have addressed my questions and I have no further comments. I feel the paper is improved and will make a nice contribution to the field.

Response to Reviews

REVIEWERS' COMMENTS:

Reviewer #1 (Remarks to the Author):

The authors have addressed my questions and I have no further comments. I feel the paper is improved and will make a nice contribution to the field.

Thank you!